# Microbiota from young mice counteracts susceptibility to age-related gout through modulating butyric acid levels in aged mice

Ning Song[1†], Hang Gao[2†], Jianhao Li[1], Yi Liu[2], Mingze Wang[1], Zhiming Ma[3], Naisheng Zhang[1*], Wenlong Zhang[1*]

[1]Department of Clinical Veterinary Medicine, College of Veterinary Medicine, Jilin University, Changchun, China; [2]Department of Bone and Joint Surgery, No 1 Hospital of Jilin University, Changchun, China; [3]Department of Gastrointestinal Nutrition and Hernia Surgery, The Second Hospital of Jilin University, Changchun, China

*For correspondence:
zhangns@jlu.edu.cn (NZ);
zwenlong123@126.com (WZ)

[†]These authors contributed equally to this work

Competing interest: The authors declare that no competing interests exist.

## eLife Assessment

This is an **important** study showing that age-related gut microbiota modulate uric acid metabolism through the NLRP3 inflammasome pathway and thereby regulate susceptibility to age-related gout. Several experimental approaches (mechanistic insights) and methods (data quality) remain **incomplete**. This article should be of interest to researchers working on gout and microbiota.

**Abstract** Gout is a prevalent form of inflammatory arthritis that occurs due to high levels of uric acid in the blood leading to the formation of urate crystals in and around the joints, particularly affecting the elderly. Recent research has provided evidence of distinct differences in the gut microbiota of patients with gout and hyperuricemia compared to healthy individuals. However, the link between gut microbiota and age-related gout remained underexplored. Our study found that gut microbiota plays a crucial role in determining susceptibility to age-related gout. Specifically, we observed that age-related gut microbiota regulated the activation of the NLRP3 inflammasome pathway and modulated uric acid metabolism. More scrutiny highlighted the positive impact of 'younger' microbiota on the gut microbiota structure of old or aged mice, enhancing butanoate metabolism and butyric acid content. Experimentation with butyrate supplementation indicated that butyric acid exerts a dual effect, inhibiting inflammation in acute gout and reducing serum uric acid levels. These insights emphasize the potential of gut microbiome rejuvenation in mitigating senile gout, unraveling the intricate dynamics between microbiota, aging, and gout. It potentially serves as a therapeutic target for senile gout-related conditions.

## Introduction

Gout, the most common inflammatory arthritis in elderly individuals, results from the deposition of monosodium urate (MSU) crystals in articular and nonarticular structures (*Dalbeth, 2016*), particularly among individuals aged 75–84 years, and the occurrence rate in this population can reach 4% (*Stamp and Jordan, 2011*). The development of gout is primarily attributed to the significant risk posed by a high serum urate concentration (*Stamp and Jordan, 2011*). Pathological hyperuricemia is defined by a serum urate concentration exceeding 408 µmol/L (6.8 mg/dL), which forms MSU crystals in vitro at physiological pH and temperature (*Chhana et al., 2015*). The reason why the elderly population

commonly experiences gout is complicated by the fact that the population is also aging, and the treatment of this disease is often intricate due to the presence of comorbidities and medications prescribed for concurrent conditions (*Stamp and Jordan, 2011*). Although the basic principles for the prevention and treatment for gout remain the same across different age groups, elderly individuals often exhibit lower tolerance for medication dosages, types, side effects, and surgical procedures due to their physiological factors. Moreover, despite the increasing severity of gout among elderly individuals, research on this issue remains scarce.

The gut microbiota plays essential roles in regulating energy and metabolism, as revealed by recent studies (*Duttaroy, 2021*; *Adak and Khan, 2019*), which have shown that individuals with hyperuricemia and gout exhibit dysbiosis of the gut microbiota (*Guo et al., 2016*; *Tiejuan et al., 2017*). Moreover, because the gut microbiota can directly participate in the metabolism of purines and uric acid (*Yamada et al., 2016*), it may play a crucial role in gout and hyperuricemia development. Although numerous studies have examined the relationship between the microbiome and early life stages, the impact of the microbiome's on aging and frailty in later life needs to be explored. Furthermore, studies have shown that the structure of the gut microbiota undergoes a gradual 'aging' process with advancing age, and this process is characterized by a decrease in the microbial diversity (*Buford, 2020*; *Lynch and Pedersen, 2016*). In addition, transplantation of a young microbiota can improve central nervous system inflammation and retinal inflammation in aged mice (*Parker et al., 2022*), and counteract age-related behavioral deficits (*Boehme et al., 2021*). We hypothesize that the high prevalence of gout in the elderly population may be closely related to its 'aging' gut microbiota. The association between the 'aging' gut microbiota and gout in elderly individuals has not been reported. Hence, we conducted a study to elucidate the influence of the aging gut microbiota on the occurrence and progression of gout in elderly individuals.

In this study, we first tested the sensitivity to MSU in different age groups and conducted a microbiota clearance (Abx) on mice of different age groups to assess their sensitivity to MSU again. At the same time, we also tested their serum uric acid (SUA) levels. We found that sensitivity to MSU increased with age, but changed after clearing the gut microbiota in terms of sensitivity to MSU and SUA levels. Then, we performed cross-age fecal microbiota transplantation (FMT) and subsequently stimulated the mice with MSU with the aim of how mice belonging to different age groups exhibit sensitivity to MSU after undergoing FMT. Because hyperuricemia is a necessary physiological factor for gout, we also evaluated the expression levels of uric acid-producing enzymes and uric acid transport proteins in the mice. Surprisingly, transplantation of the gut microbiota from aged mice into young mice significantly increased their sensitivity to MSU. Conversely, the transplantation of the gut microbiota from young mice into aged mice significantly decreased their sensitivity to MSU. These findings suggest that the gut microbiota of older individuals plays a promoting role in the occurrence and progression of gout. Moreover, an analysis of the SUA levels of the mice after cross-age FMT yielded similar results. To investigate the underlying mechanisms of how gut microbiota influences gout and hyperuricemia, we performed 16S rDNA sequencing and untargeted metabolomics analysis of fecal samples. We then observed a significant increase in the abundance of *Bifidobacterium* and *Akkermansia* in the gut microbiota of young mice and old or aged mice after transplantation of the gut microbiota from young mice. To further investigate the potential biological pathways affected by the gut microbiota, we performed functional metagenomic analysis using Tax4FUN software and found that butanoate metabolism was more robust in young mice than in aged mice. Furthermore, we also observed that transplantation of the gut microbiota from young mice to aged mice enhanced the butanoate metabolism of the recipient mice. Due to the limitations of untargeted metabolomics, we did not observe any differences in the levels of butyric acid among the different groups. However, the pathway data obtained by fecal untargeted metabolomics also yielded similar results. Based on the above-described results, we hypothesize that butyrate may play a significant role in these processes. Excitingly, the results from a short-chain fatty acid (SCFA) analysis support our hypothesis. Furthermore, a supplementation experiment using butyrate revealed that the results aligned well with the cross-age FMT findings, which suggests that transplantation of the gut microbiota from young mice into aged mice can effectively prevent gout and hyperuricemia, and that butyrate is likely the critical factor playing a crucial role. Our research findings demonstrate the potential of young gut microbiota in preventing gout in elderly individuals and provide new insights and perspectives for the prevention and treatment of gout in elderly individuals.

## Results

### Gout susceptibility increases with age, related to gut microbiota

In this study, we used the male C57BL/6 mice belonging to three age groups: young (~3 months), old (~18 months), and aged (~24 months) mice (*Figure 1a*). To investigate the impact of age on gout, we used a mouse model in which subcutaneous injections of MSU crystals were administered to the dorsal aspect of the hind paws (*Schauer et al., 2014*; *Yang et al., 2018*; *Mariotte et al., 2020*). The result showed that the old group exhibited a significant increase in footpad swelling compared with the young group, and the levels of IL-1β were significantly elevated in the old and aged groups (*Figure 1b and c*). The level of IL-6 was significantly elevated in the old and aged groups after MSU was stimulated. A prominent correlation was found between the occurrence of gout and the concentration of SUA, which exhibits an upward trend with advancing age (*Stamp and Jordan, 2011*). Thus, we also measured the SUA level and found uric acid levels significantly increase with age (*Figure 1d*). There are reports indicating a close relationship between gut microbiota and gout (*Guo et al., 2016*). Therefore, we simultaneously tested the gut microbiota of mice at different ages. The results showed that as age increases the amplicon sequence variants (ASVs) showed a declining trend (*Figure 1e*). Moreover, the principal coordinates analysis (PCoA) results revealed distinctive differences in the phylogenetic community structures between these groups (*Figure 1f*). Then, we carried out antibiotics (ABX) cocktail on mice of different age groups, found changes in the footpad swelling, IL-1β, and discovered no differences in the level of SUA (*Figure 1g–i*). These data highlight that aging-associated changes in the gut microbiota exacerbate gout attacks.

### Aged-to-young FMT worsens acute gout, whereas young-to-aged FMT reduced this disease

Based on the above results, we conducted fecal microbiota exchanges between male C57BL/6 mice also belonging to three age groups: young (~3 months), old (~18 months), and aged (~24 months) mice. The experimental design and timeline are presented in *Figure 2a*. To investigate the impact of cross-age FMT on gout, we used a gout mouse model as aforementioned. The results of *Figure 2—figure supplement 1a* showed that young mice transplanted with fecal microbiota from mice in the old or aged group (Young + Old or Young + Aged) exhibited a significant increase in footpad swelling compared with the control group (Young + PBS). However, no significant difference in footpad swelling was found between old or aged mice transplanted with fecal microbiota from mice in the young groups (Old + Young and Aged + Young) and the control groups (Old + PBS and Aged + PBS) (*Figure 2—figure supplement 1*). Subsequently, we conducted a comparative analysis of H&E-stained tissue sections (scar, 1000 μm), derived from mice with cross-age FMT and MSU-induced acute gout. We found that the Young + Old and Young + Aged group showed more inflammatory cell infiltration in the subcutaneous soft tissues than the Young + PBS group, whereas the Old + Young and Aged + Young groups showed less inflammatory cell infiltration (*Figure 2b*). The inflammatory factors in the foot tissue of the mice with cross-age FMT were then examined. The levels of IL-1β, IL-6, and TNFα were significantly elevated in the Young + Aged group and significantly lower in the Aged + Young group compared with the Aged + PBS group (*Figure 2c–e*). Meanwhile, to investigate the impact of cross-age FMT on MSU-induced inflammation, we utilized an animal model of C57BL/6 mice administered an intraperitoneal injection of 2.5 mg of MSU (*Goldberg et al., 2017*). Then, 6 hr after MSU intraperitoneal injection, we washed the mouse peritoneal cavity with sterile PBS and collected the peritoneal fluid and peritoneal cells separately. Simultaneously, we also examined the levels of inflammatory cytokines in mouse serum. Consistent with the acute gout model, the Young + Old or Young + Aged group exhibited more pronounced activation of IL-1β, IL-6, and TNF-α in peritoneal fluid (*Figure 2f–h*) and serum (*Figure 2—figure supplement 1b–d*). Although no differences in the IL-6 levels of serum and peritoneal fluid were found between the Old + PBS and Old + Young groups or between the Aged + Young and Aged + PBS groups (*Figure 2—figure supplement 1c*, *Figure 2g*), a significant reduction in IL-1β and TNF-α was observed (*Figure 2—figure supplement 1b and d*, *Figure 2f and h*). These findings suggest that the increased susceptibility to gout in the elderly may closely related to 'aging' gut microbiota.

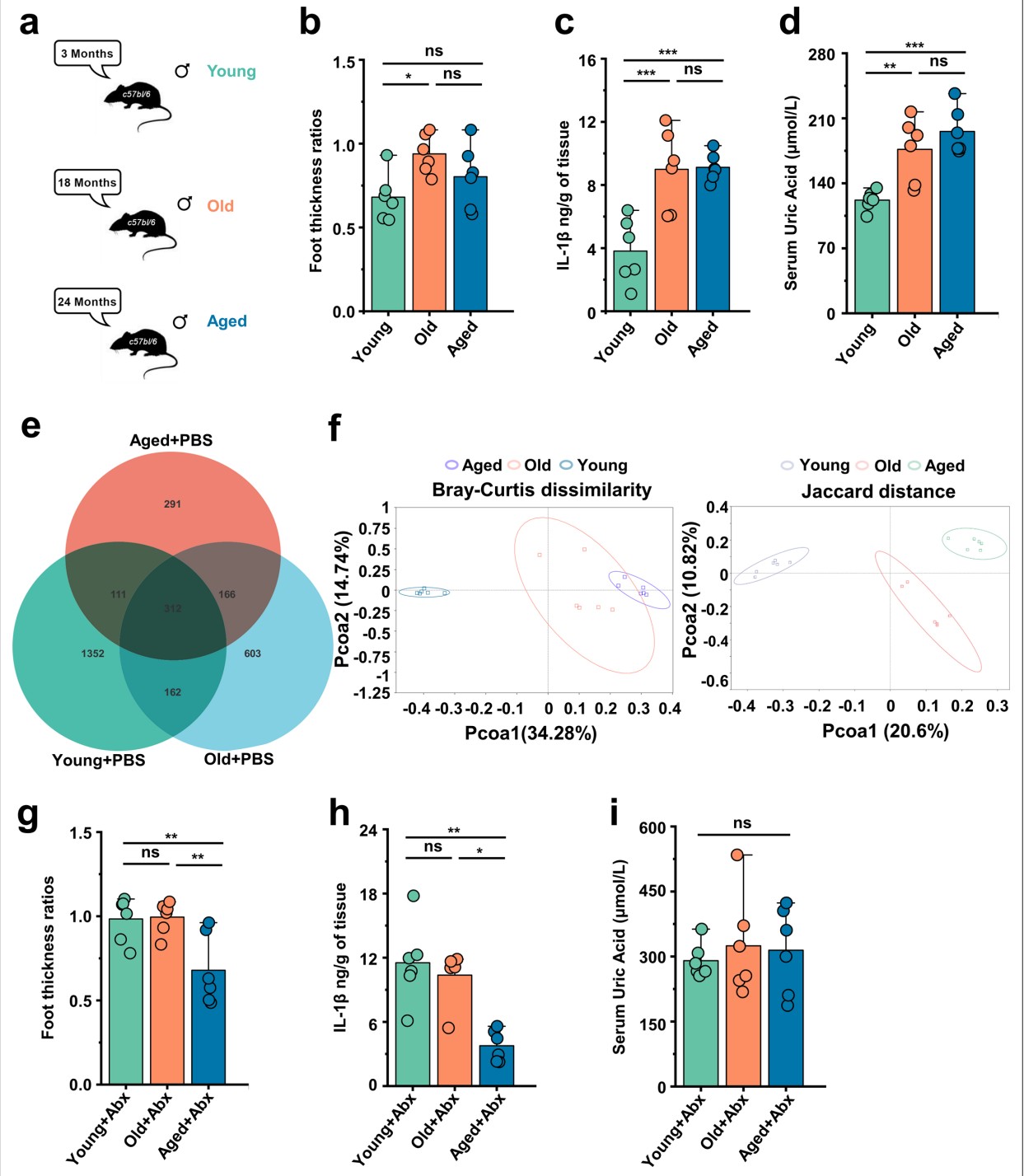

**Figure 1.** Gout susceptibility increases with age, related to gut microbiota. (**a**) Mice of different ages. (**b–d**) The foot thickness ratios (**b**), foot tissue's IL-1β concentrations (**c**), and serum concentrations of uric acid of three different age ranges groups (**d**) were tested after monosodium urate (MSU) administration (n = 6). (**e, f**) The three different age ranges groups' amplicon sequence variants (ASVs) (**e**) and principal coordinates analysis (PCoA) (using Bray–Curtis dissimilarity and Jaccard distance) (**f**). (**g–i**) The foot thickness ratios (**g**), foot tissue's IL-1β concentrations (**h**), and serum concentrations of uric acid of three different age ranges groups (**i**) (treated with antibiotics [ABX] cocktail) were tested after MSU administration (n = 6). Values are presented as the mean ± SEM. Differences were assessed by one-way ANOVA and denoted as follows: *p<0.05, **p<0.01, and ***p<0.001, 'ns' indicates no significant difference between groups.

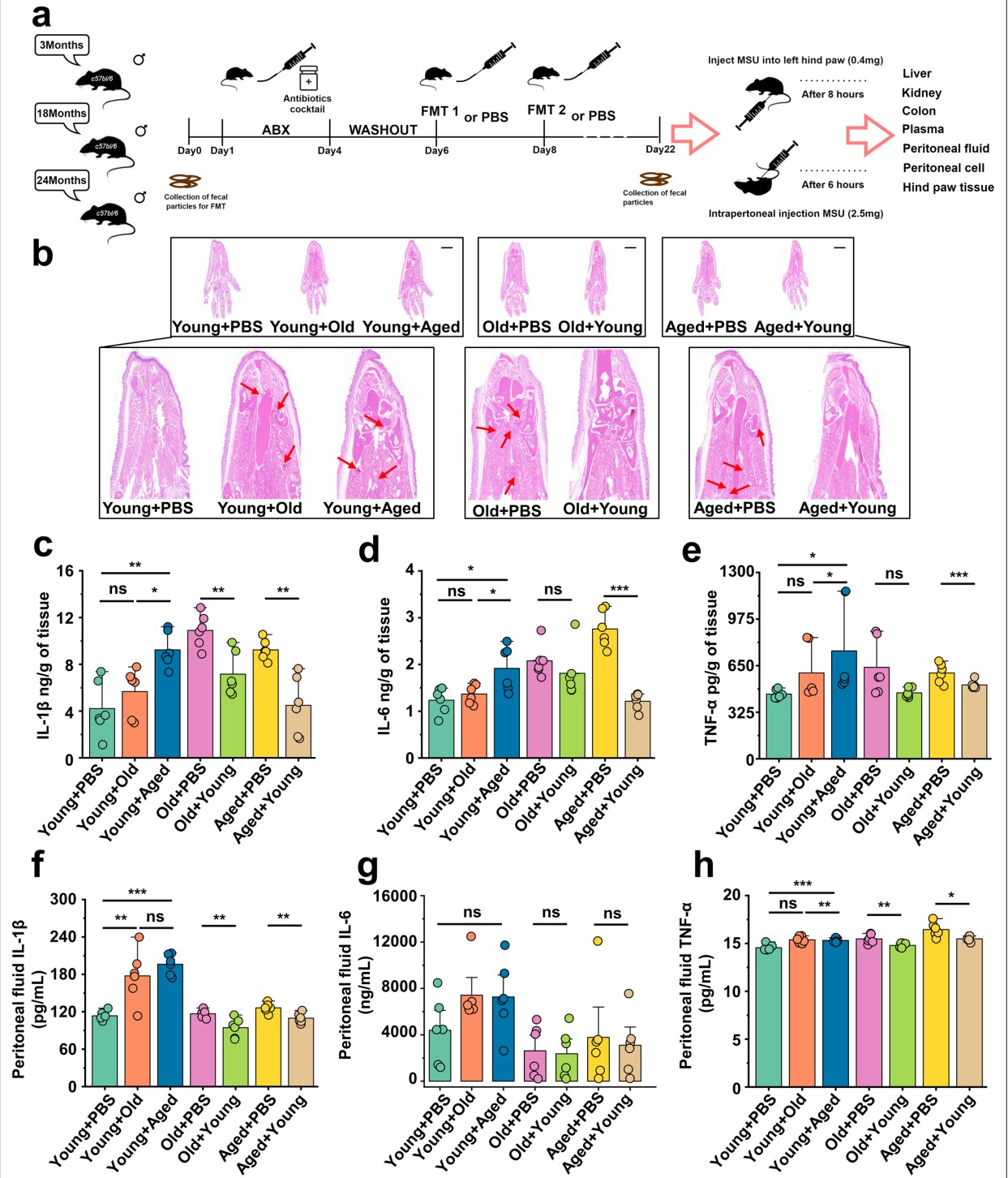

**Figure 2.** Aged-to-young fecal microbiota transplantation (FMT) worsens acute gout, whereas young-to-aged FMT reduced this disease. (**a**) Overall experimental design and timeline for experiments. (**b**) Representative H&E-stained images of left foot tissues. Scale bars 1000 μm and ×3× magnification. (**c–e**) Foot tissue inflammatory parameters, including IL-1β (**c**), IL-6 (**d**), and TNF-α (**e**) concentrations, from the indicated mice are shown (n = 6). (**f–h**) The peritoneal fluid concentrations of IL-1β (**f**), IL-6 (**g**), and TNF-α (**h**) inflammatory parameters were measured in the indicated mice (n = 6). Values are presented as the mean ± SEM. Differences were assessed by *t*-test or one-way ANOVA and denoted as follows: *p<0.05, **p<0.01, and ***p<0.001, 'ns' indicates no significant difference between groups.

The online version of this article includes the following figure supplement(s) for figure 2:

**Figure supplement 1.** Young-to-aged FMT reduced the production of blood inflammatory factors stimulated by MSU.

## 'Younger' gut microbiota suppresses NLRP3 inflammasome pathway, 'aging' gut microbiota promotes

The pathogenesis of acute gout has been primarily linked to the activation of proinflammatory pathways, notably NLRP3, and this activation instigates a surge in the production of the inflammatory cytokines, which further underscores their pivotal roles in the progression of the diseases (*Martinon et al., 2006*). We primarily examined proteins of foot tissue and peritoneal cells associated with the NLRP3 inflammasome pathway. Interestingly, we discovered that the Young + Aged group demonstrated more pronounced activation of NLRP3, pro-caspase-1, caspase-1, and IL-1β compared with the control group (*Figure 3a and b*). However, the Old + Young and Aged + Young groups showed lower protein levels of caspase-1 and IL-1β compared with the control group (*Figure 3c–f*), which suggests that the gut microbiota of elderly mice may make them more sensitive to MSU stimulation, whereas the gut microbiota of young mice can effectively resist the MSU stimulation and inhibit caspase-1 cleavage and IL-1β secretion. Meanwhile, the NLRP3 inflammasome pathway also was investigated in peritoneal cells, and similar results were obtained in the gout model. The fecal microbiota from mice in the old or aged group exacerbated the activation of pro-caspase-1, caspase-1, pro-IL-1β, and IL-1β in peritoneal cells, increasing the production of IL-1β (*Figure 3—figure supplement 1a and b*). In contrast, the fecal microbiota from mice in the young group inhibited the cleavage of caspase-1 and the secretion of IL-1β (*Figure 3—figure supplement 1c–f*).

## Beneficial effects of FMT from young to aged mice on uric acid metabolism

Serum samples were collected from the different groups, and their SUA levels were initially assessed. Surprisingly, we observed an elevation in the average SUA levels in the Young + Old or Young + Aged group, whereas the Old + Young or Aged + Young group exhibited decreases in the SUA levels compared with the same-age control group (*Figure 4a*). Our study showed that compared with those of the same-age control group, the serum levels of aspartate aminotransferase (AST) and alanine aminotransferase (ALT) were elevated in the Young + Old and Young + Aged groups, although the differences were not significant, whereas the Aged + Young groups exhibited modest decreases in these levels (*Figure 4—figure supplement 1a and b*). Furthermore, similar trends were found for indicators (Crea and BUN) of renal function (*Figure 4—figure supplement 1c and d*). Hyperuricemia is attributed to increased uric acid synthesis and decreased uric acid excretion in the body. We first measured the activity of enzymes involved in uric acid synthesis, namely adenosine deaminase (ADA), guanine deaminase (GDA), and xanthine dehydrogenase (XOD), in the liver and that of (XOD) in the kidney. The activity of ADA, GDA, and XOD in the liver, and the activity of XOD in the kidney of the Young + Old and Young + Aged groups did not significantly differ from those of the same-age control groups (*Figure 4b–e*). The fecal microbiota from mice in the young group induced a notable decrease in the activity of enzymes related to uric acid synthesis, and the most prominent reductions were found for ADA and XOD activity in the liver and XOD activity in the kidney (*Figure 4b, d and e*). We then examined the mRNA levels of relevant proteins involved in uric acid transport. We initially measured the renal injury marker *Havcr1* (KIM-1) and observed that the fecal microbiota from mice in the young group contributed to attenuating the mRNA expression levels of this marker compared with the same-age control group (*Figure 4f*). The urate transporter *Slc22a12* (URAT1), which is responsible for uric acid reabsorption, exhibited a similar trend (*Figure 4g*). However, cross-age FMT did not significantly impact the mRNA expression of another uric acid reabsorption protein, *Slc2a9* (GLUT9) (*Figure 4h*). The Young + Aged group showed lower mRNA expression levels of the uric acid excretion proteins *Slc22a6* (OAT1) and *Slc22a8* (OAT3) compared with the Young + PBS group (*Figure 4i and j*). Although the mRNA expression levels of Slc22a6 (OAT1) and Slc22a8 (OAT3) did not show significant differences, a significant increasing trend was found for the other uric acid excretion protein, Abcg2 (ABCG2), in aged mice transplanted with fecal microbiota from mice in the young group exhibited a significant increasing trend (*Figure 4k*). Because Abcg2 (ABCG2) is also expressed in the intestine, we examined its mRNA expression levels in the colon and found that the fecal microbiota from mice in the aged group could inhibit its expression (*Figure 4—figure supplement 1e*). Similar trends were found for the mRNA expression levels of mice in *Tjp1* (ZO-1) and *F11r* (JAMA) in the colon of the Young + Aged group (*Figure 4—figure supplement 1g and h*). No significant alterations in the colonic Ocln (Occludin) mRNA expression levels were observed among all the groups (*Figure 4—figure*

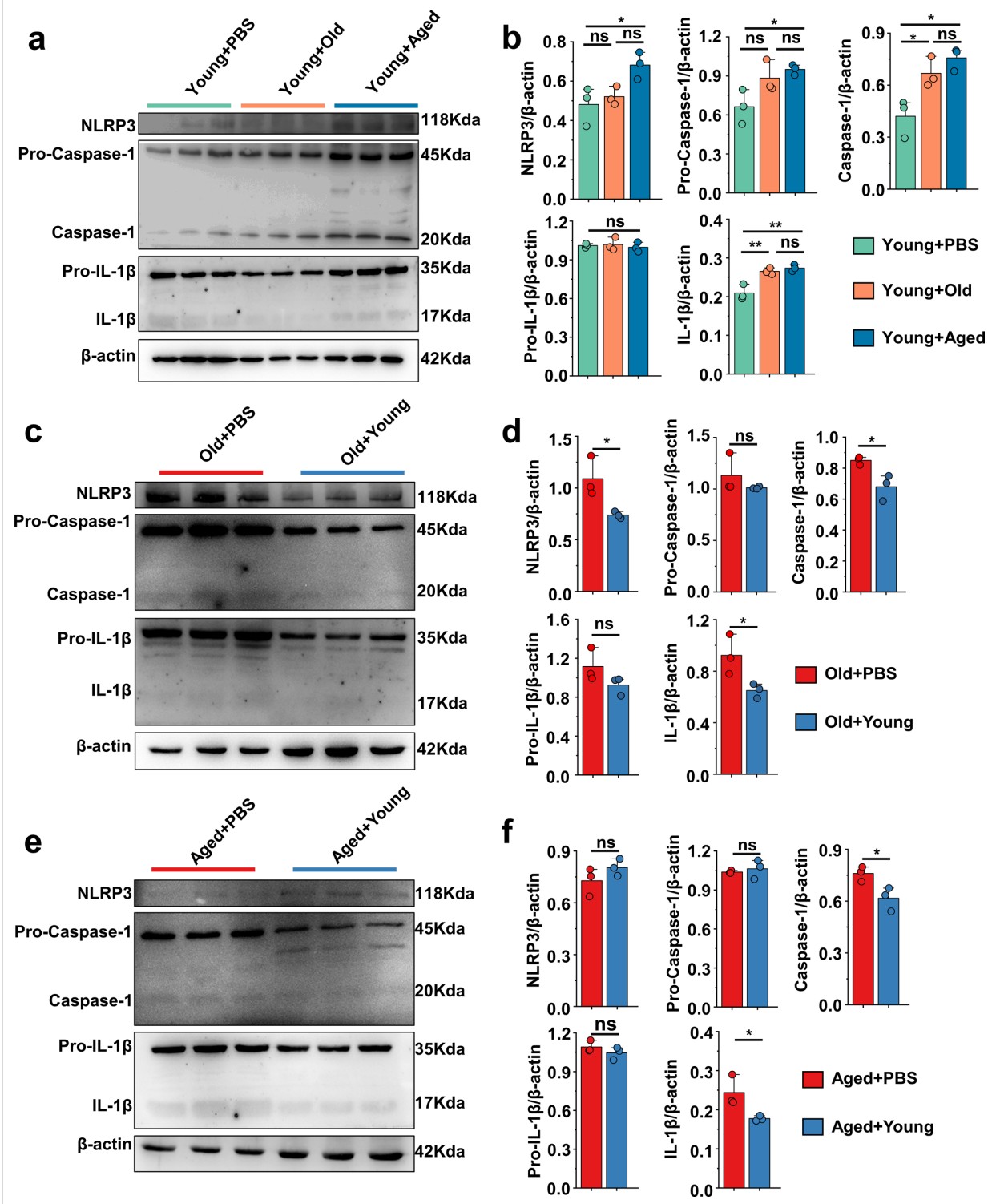

**Figure 3.** 'Younger' gut microbiota suppresses NLRP3 inflammasome pathway, 'aging' gut microbiota promotes. (**a, b**) Representative western blot images and band density (Young + PBS, Young + Old, and Young + Aged) of foot tissue NLRP3 pathways proteins (n = 3). (**c, d**) Representative western blot images and band density (Old + PBS and Old +Young) of foot tissue NLRP3 pathways proteins (n = 3). (**e, f**) Representative western blot images and band density (Aged + PBS and Aged + Young) of foot tissue NLRP3 pathways proteins (n = 3). Values are presented as the mean ± SEM. Differences were assessed by *t*-test or one-way ANOVA and denoted as follows: *p<0.05, **p<0.01, and ***p<0.001, 'ns' indicates no significant difference between groups.

The online version of this article includes the following source data and figure supplement(s) for figure 3:

*Figure 3 continued on next page*

*Figure 3 continued*

**Source data 1.** Original Western blot image of the NLRP3 inflammasome pathway in foot tissue.

**Source data 2.** Original Western blot image of the NLRP3 inflammasome pathway in foot tissue.

**Figure supplement 1.** Young gut microbiota suppresses the NLRP3 inflammasome pathway in peritoneal cells, whereas aging gut microbiota activates it.

**Figure supplement 1—source data 1.** Original Western blot image of the NLRP3 inflammasome pathway in peritoneal cells.

**Figure supplement 1—source data 2.** Original Western blot image of the NLRP3 inflammasome pathway in peritoneal cells.

*supplement 1f*). However, the mRNA expression levels of F11r (JAMA) in the Old + Young and Aged + Young groups were significantly different from those in the same-age control groups (*Figure 4—figure supplement 1h*). These results suggest that the fecal microbiota from mice in the old or aged group leads to insufficient uric acid excretion, whereas the fecal microbiota from mice in the young group promotes uric acid elimination, inhibits reabsorption, and may contribute to the integrity of the intestinal barrier structure and the maintenance of normal physiological function.

## Modifications in the gut microbiota composition following cross-age FMT

To characterize the age-related alterations in the gut microbiota and assess changes following transplantation, we conducted 16S rDNA amplicon sequencing of all the mice at the endpoint. We focused on investigating the gut microbiota of mice after cross-age FMT. We detected the relative abundance of the top 10 phylum levels in each group (*Figure 5a*) and found no significant differences in the abundance of *Bacteroidetes* and *Firmicutes* in the transplanted groups compared with that of their respective control groups (*Figure 5—figure supplement 1a and b*). No significant differences were found in the ratio of *Firmicutes* to *Bacteroidetes* (*Figure 5—figure supplement 1c*). The species richness, evenness, and rarity are fundamental components of biodiversity and are commonly quantified using indices such as Chao1, observed otus, Shannon, and Simpson indices. Although no significant differences in the Simpson index were found among these groups (*Figure 5—figure supplement 1e*), and the indices of the Old + Young and Aged + Young groups were not significantly different from those of the Old + PBS and Aged + PBS groups, respectively, the Chao 1 (*Figure 5b*), observed_otus (*Figure 5c*), and Shannon (*Figure 5—figure supplement 1d*) indices of the Young +Aged group were significantly lower than those of the Young + PBS group. These findings are in line with other studies (*Ghosh et al., 2022*), indicating declines in the richness and diversity of the gut microbiota during aging. Moreover, the PCoA results revealed distinctive differences in the phylogenetic community structures between these groups. We showed that the Young + Old and Young + Aged groups tended to be closer to the Old + PBS and Aged + PBS groups, and the Old + Young and Aged + Young groups tended to be closer to the Young + PBS group (*Figure 5d*). We then compared the abundance of the top 15 genus level among all the groups (*Figure 5—figure supplement 1f*). A Metastats analysis found that the abundance of *Bifidobacterium* significantly differed between the Young + PBS group and the Young + Old and Young + Aged groups, and that the abundance of *Lachnoclostridium* showed an increasing trend in the Old + Young and Aged + Young groups (*Figure 5e*). The ternary plot showed that *Akkermansia* appears to be the dominant species in the Young + PBS, Old + Young, and Aged + Young group (*Figure 5f*). According to previous reports (*Sarao and Arora, 2017*; *Nowak et al., 2019*; *Derrien et al., 2004*; *Zhang et al., 2019*) on *Bifidobacterium* and *Akkermansia*, we hypothesize that these genera or their metabolites may play a key role in resistance to gout and hyperuricemia. Based on recent research findings (*Lee et al., 2020*) and a KEGG analysis of bacterial community functions, we discovered that butanoate metabolism was more robust in the Young + PBS group than in the Old + PBS and Aged + PBS groups (*Supplementary file 2a and b*). However, no similar phenomenon was observed in the Young + Old and Young + Aged groups compared with the Young + PBS group (*Supplementary file 2c and d*). More interestingly, the Old + Young and Aged + Young groups showed stronger butanoate metabolism than the Old + PBS and Aged + PBS groups, respectively (*Supplementary file 2e and f*). Considering the results from the analysis modifications in the gut microbiota composition after cross-age FMT, both *Bifidobacterium* and *Akkermansia* metabolites include SCFAs. These findings indicate that butyric acid derived from the young microbiome may be the critical element responsible for controlling gout. Numerous studies have reported the

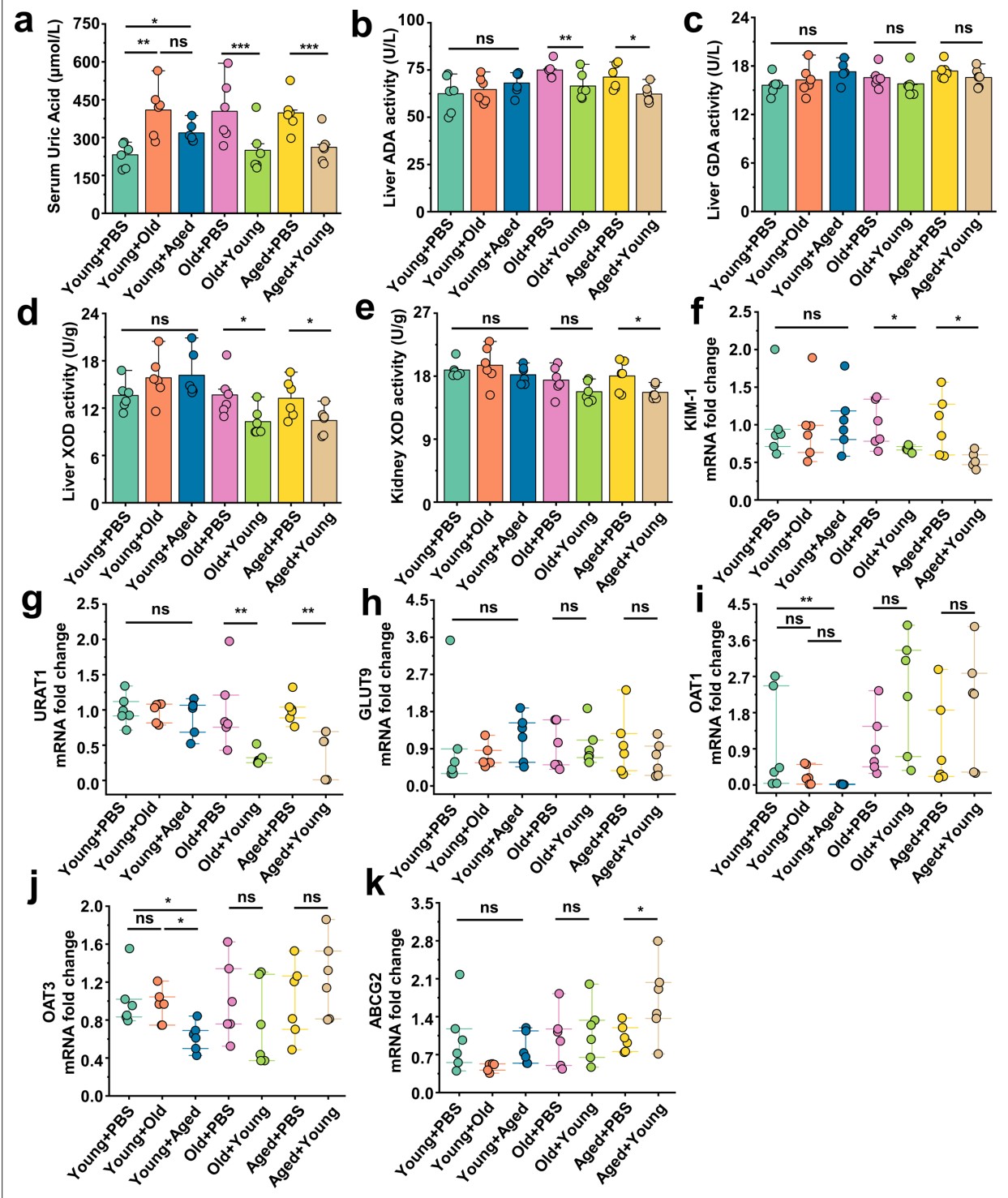

**Figure 4.** Beneficial effects of fecal microbiota transplantation (FMT) from young to aged mice on uric acid metabolism. (**a**) All groups' serum concentrations of uric acid (n = 6). (**b–d**) The activity of uric acid-producing enzymes of liver in the cross-age FMT group and its control group (n = 6), including adenosine deaminase (ADA) (**b**), guanine deaminase (GDA) (**c**), and xanthine dehydrogenase (XOD) (**d**). (**e**) The activity of XOD of kidney in the cross-age FMT group and its control group (n = 6). (**f**) Relative kidney injury molecule-1 (KIM-1) expression in the indicated groups by qPCR (n = 6). (**g, h**) Relative renal genes for uric acid reabsorption expression in the indicated groups by qPCR (n = 6), including URAT1 (**g**) and GLUT9 (**h**). (**i–k**) Relative renal genes for uric acid excretion expression in the indicated groups by qPCR (n = 6), including OAT1 (**i**), OAT3 (**j**), and ABCG2 (**k**). Values are presented as the mean ± SEM. Differences were assessed by *t*-test or one-way ANOVA and denoted as follows: *p<0.05, **p<0.01, and ***p<0.001, 'ns' indicates no significant difference between groups.

*Figure 4 continued on next page*

*Figure 4 continued*

The online version of this article includes the following figure supplement(s) for figure 4:

**Figure supplement 1.** Young gut microbiota improves hepatic, renal, and intestinal functions in aged mice.

beneficial effects of butyric acid, but further research on the relationship between butyric acid and gout is needed.

## Changes in fecal microbiota metabolism and pathways after cross-age FMT

We also performed an untargeted metabolomics analysis of samples collected after cross-age FMT to investigate the research question. Chemical classification of the metabolites identified in this study was performed, and a pie chart was generated to reflect the distribution and numbers of metabolites in each category. The pie chart for class I metabolite classification and KEGG pathway annotation is shown in *Figure 6—figure supplement 1a*. Among the 42 samples analyzed in this study, a total of 1009 metabolites were identified in the positive ion mode and 522 metabolites were identified in the negative ion mode. The volcano plot provides an overview of the distribution of differentially expressed metabolites (*Figure 6—figure supplement 1b*). We also used the KEGG database for metabolic analysis and network research of the identified biological entities. The enrichment results are based on KEGG pathway units, and hypergeometric testing was performed to identify the pathways enriched in differentially abundant metabolites compared with all specified metabolite backgrounds. Through pathway enrichment, we discerned and elucidated the principal biochemical metabolic pathways and signal transduction pathways in which the differentially abundant metabolites actively participate. The metabolites showing differential abundance between the Young + PBS and Old + PBS groups were enriched in the secondary bile acid biosynthesis pathway (*Figure 6—figure supplement 2a*). Interestingly, the metabolites showing differential abundance between the Young + PBS and Aged + PBS groups were also enriched in the butanoate metabolism pathway (*Figure 6—figure supplement 2b*). The same phenomenon was found from the comparisons of the Young + PBS and Young + Old groups, Young + PBS and Young + Aged groups, Old + PBS and Old + Young groups, and Aged + Young and Aged + PBS groups: the metabolites identified as showing differential abundance between the groups metabolites were enriched in the butanoate metabolism pathway (*Figure 6a–d*). Furthermore, we found that the metabolites showing differential abundance between the Old + PBS and Old + Young groups and between the Aged + Young and Aged + PBS groups were enriched in the secondary bile acid biosynthesis pathway (*Figure 6c and d*). Due to the limitations of untargeted metabolomics, we did not observe any differences in butyric acid. However, the results were consistent with the predicted functional capabilities of microbial communities and corresponded to butanoate metabolism. Based on the abovementioned results, butyrate may play an important role.

## Butyrate supplementation inhibits gout and MSU-induced peritonitis

Based on the predictions of the functional capabilities of microbial communities and untargeted metabolomics results, we hypothesized that butyrate might play a key role. Because butyric acid is an SCFA, we measured the levels of SCFAs (nk/g) in the Young + PBS, Aged + PBS, and Aged + Young groups. Subsequently, we observed that the butyric acid content in the feces of Young + PBS group was significantly higher than that in the Aged + PBS group. Simultaneously, the butyric acid content in the feces of mice in the Aged + Young was also significantly higher than that in the Aged + PBS (*Figure 7a*). 2-Methylvalerate and 3-methylvalerate were not detected, and 4-methylvaleric was detected in only a subset of samples, thus, these results are not presented. The difference analysis of the detected SCFAs is shown in *Figure 7—figure supplement 1a–h*. Therefore, mice were supplemented with butyrate for 14 days and then subjected to intraperitoneal injections of MSU into the dorsal aspect of the hind paws to observe its anti-inflammatory effects in acute gout. The experimental results showed that butyrate exerted significant anti-inflammatory effects in the acute gout model with inflammation induced by intraperitoneal injection of MSU. Mice with gout model supplemented with butyrate showed significant reductions in the foot thickness ratio (*Figure 7b*) and the levels of inflammatory factors (IL-1β, IL-6, and TNF-α) in their foot tissue (*Figure 7c–e*). Pathological sections revealed that the foot tissue of mice supplemented with butyrate exhibited less inflammatory cell infiltration

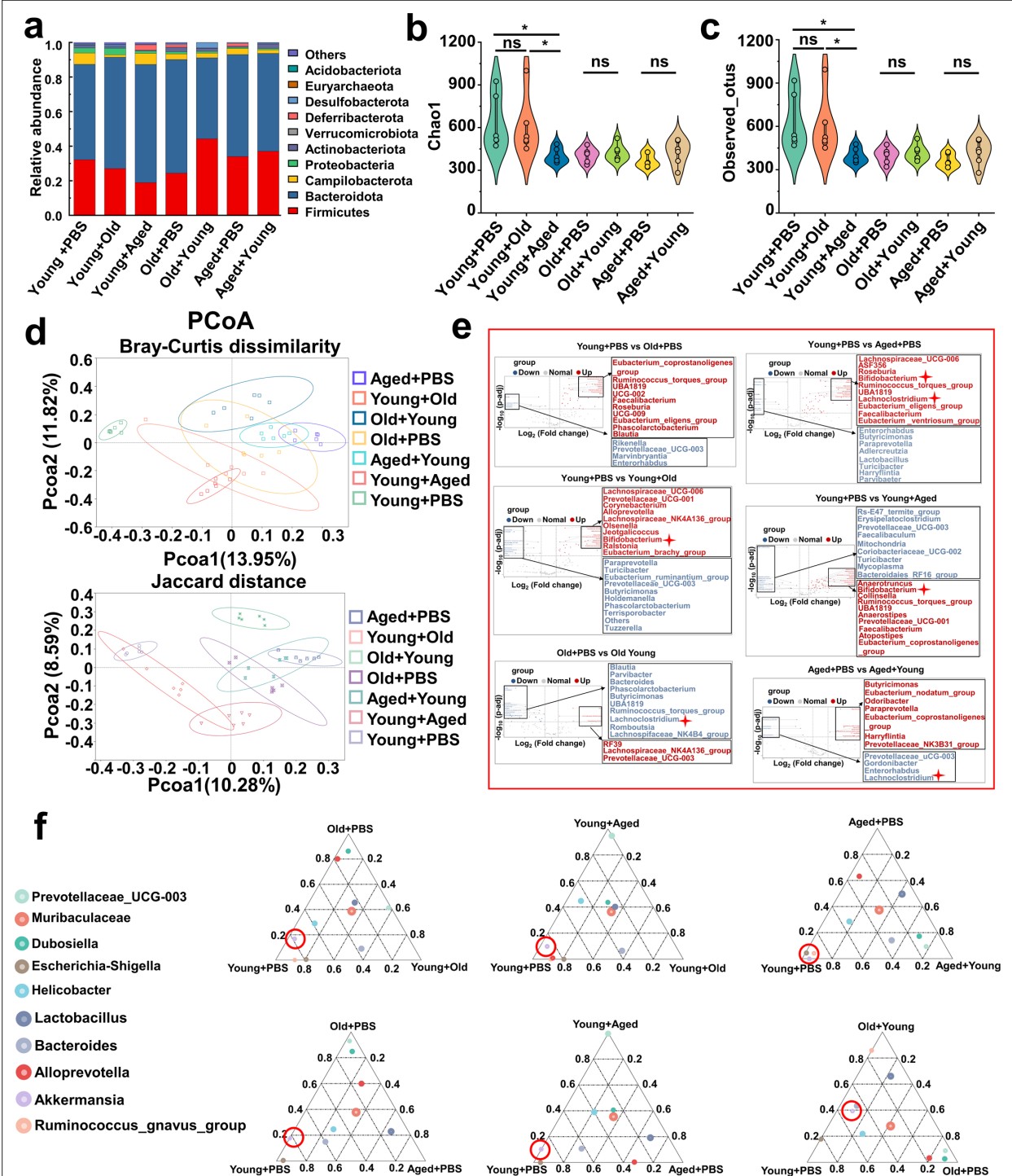

**Figure 5.** Modifications in the gut microbiota composition following cross-age fecal microbiota transplantation (FMT). (**a**) Bacterial composition at the phylum levels (top 10) of the indicated groups (n = 6). (**b, c**) Alpha diversity indices including Chao1 (**b**) and observed_otus (**c**) index in the indicated groups (n = 6). (**d**) β-diversity difference among the seven groups analyzed by the principal coordinates analysis (PCoA) using Bray–Curtis dissimilarity and Jaccard distance. (**e**) Volcano plot of inter-group significance analysis using metastat (t-test, p<0.05). (**f**) The ternary plot of three different groups among the seven groups at genus levels (top 10).

The online version of this article includes the following figure supplement(s) for figure 5:

**Figure supplement 1.** Cross-age fecal microbiota transplantation (FMT) dynamically alters gut microbiota architecture.

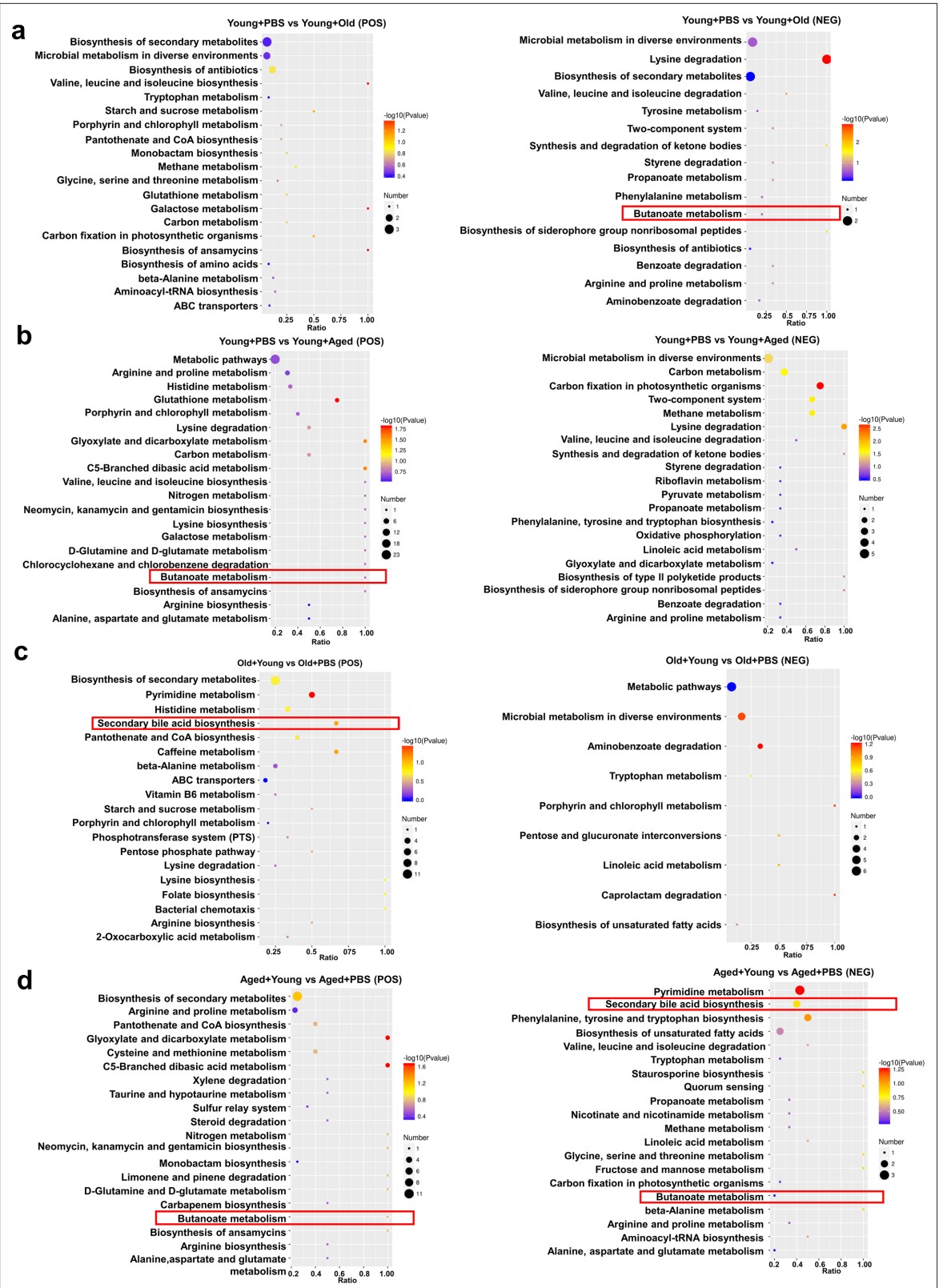

**Figure 6.** Changes in fecal microbiota metabolism and pathways after cross-age fecal microbiota transplantation (FMT). (**a**) Comparison of differential KEGG enrichment bubble plots between Young + PBS and Young + Old. (**b**) Comparison of differential KEGG enrichment bubble plots between Young + PBS and Young + Aged. (**c**) Comparison of differential KEGG enrichment bubble plots between Old + PBS and Old + Young. (**d**) Comparison of differential KEGG enrichment bubble plots between Aged + PBS and Aged + Young. The enrichment analysis was performed at the KEGG pathway

*Figure 6 continued on next page*

*Figure 6 continued*

level using a hypergeometric test, as shown in the figure below. The pathways that were significantly enriched in the differential metabolites compared to the background of all identified metabolites. Pathway enrichment analysis enables us to determine the major biochemical metabolic pathways and signaling transduction pathways that are implicated by the differential metabolites.

The online version of this article includes the following figure supplement(s) for figure 6:

**Figure supplement 1.** Metabolite classification and differential metabolites in each group.

**Figure supplement 2.** Comparison of KEGG enrichment bubble plots between different age groups.

than that of the control group (*Figure 7f*). Moreover, after supplementation with butyrate, mice that received an intraperitoneal injection of MSU showed significant decreases in the levels of inflammatory factors (IL-1β, IL-6, and TNF-α) in both serum (*Figure 7—figure supplement 1i–k*) and peritoneal fluid (*Figure 7g–i*) compared with those in the control group. Subsequently, we also analyzed the expression of proteins associated with the NLRP3 inflammasome pathway. Although no significant change in the expression of NLRP3 after supplementation with butyrate was observed in both the gout and peritonitis models, supplementation with butyrate effectively suppressed the production of caspase-1 and IL-1β (*Figure 7j*, *Figure 7—figure supplement 1l*). The anti-inflammatory effects of butyrate have been widely reported, but limited studies have investigated its specific on gout. Here we demonstrate that butyrate effectively inhibits inflammation induced by MSU stimulation.

## Serum uric acid-lowering effect of butyrate in old or aged mice

This has been proved in *Figure 1d* that the SUA levels in mice of different age groups exhibited increase in their SUA levels with increase age. There is scarce research on the impact of butyrate on SUA levels, and the effects of butyrate on hepatic uric acid production or renal uric acid excretion have not been investigated. Additionally, the impact of butyrate on the SUA levels in aged mice remains unexplored. Aged mice were supplemented with butyrate for 14 days, and we found that butyrate supplementation significantly reduced the SUA levels in 18-month-old and 24-month-old mice (*Figure 8a*). In this regard, we also assessed the activity of enzymes implicated in uric acid synthesis, namely, ADA, GDA, and XOD in the liver, as well as that of XOD in the kidney. In addition to identifying that butyrate can reduce the activity of ADA in 24-month-old mice (Aged) (*Figure 8b*), and supplementation with butyrate did not have significant effects on the activities of enzymes involved in uric acid production in both 18-month-old and 24-month-old mice (*Figure 8c–e*). Butyrate may not necessarily lower the SUA levels by inhibiting uric acid production. Because butyrate might exert its influence on SUA levels by affecting uric acid transport, we subsequently examined the expression of relevant transporter mRNA in mouse kidneys and the expression of the colonic Abcg2 (ABCG2) transporter that facilitates uric acid excretion. The expression of the renal injury marker *Havcr1* (KIM-1) was significantly decreased in mice supplemented with butyrate (*Figure 8f*). In contrast to the results obtained cross-age FMT, no significant difference in the expression of Slc22a12 (URAT1) was observed (*Figure 8g*). However, the expression of Slc2a9 (GLUT9) was significantly decreased in 24-month-old mice after butyrate supplementation, whereas no significant difference was observed in the 18-month-old mice (*Figure 8h*). We also found that the supplementation of old or aged mice with butyrate resulted in an increased mRNA expression level of uric acid transporters involved in uric acid excretion and significantly increased the expression of Slc22a6 (OAT1) and Slc22a8 (OAT3) (*Figure 8i and j*). Although no significant difference in the expression of Abcg2 (ABCG2) was detected in old mice, a significant increased in its expression was observed in aged mice after supplementation with butyrate (*Figure 8k*). Meanwhile, the mRNA expression of Abcg2 (ABCG2) in the colons was significantly increased in old or aged mice after supplementation with butyrate (*Figure 8l*). Moreover, supplementation with butyrate significantly increased the mRNA expression of *Tjp1* (ZO-1), Ocln (Occludin), and F11r (JAMA) in the colons of mice (*Figure 8—figure supplement 1a–c*). Based on the abovementioned results, we found that supplementation with butyrate promotes uric acid excretion and improves the intestinal tight junction integrity.

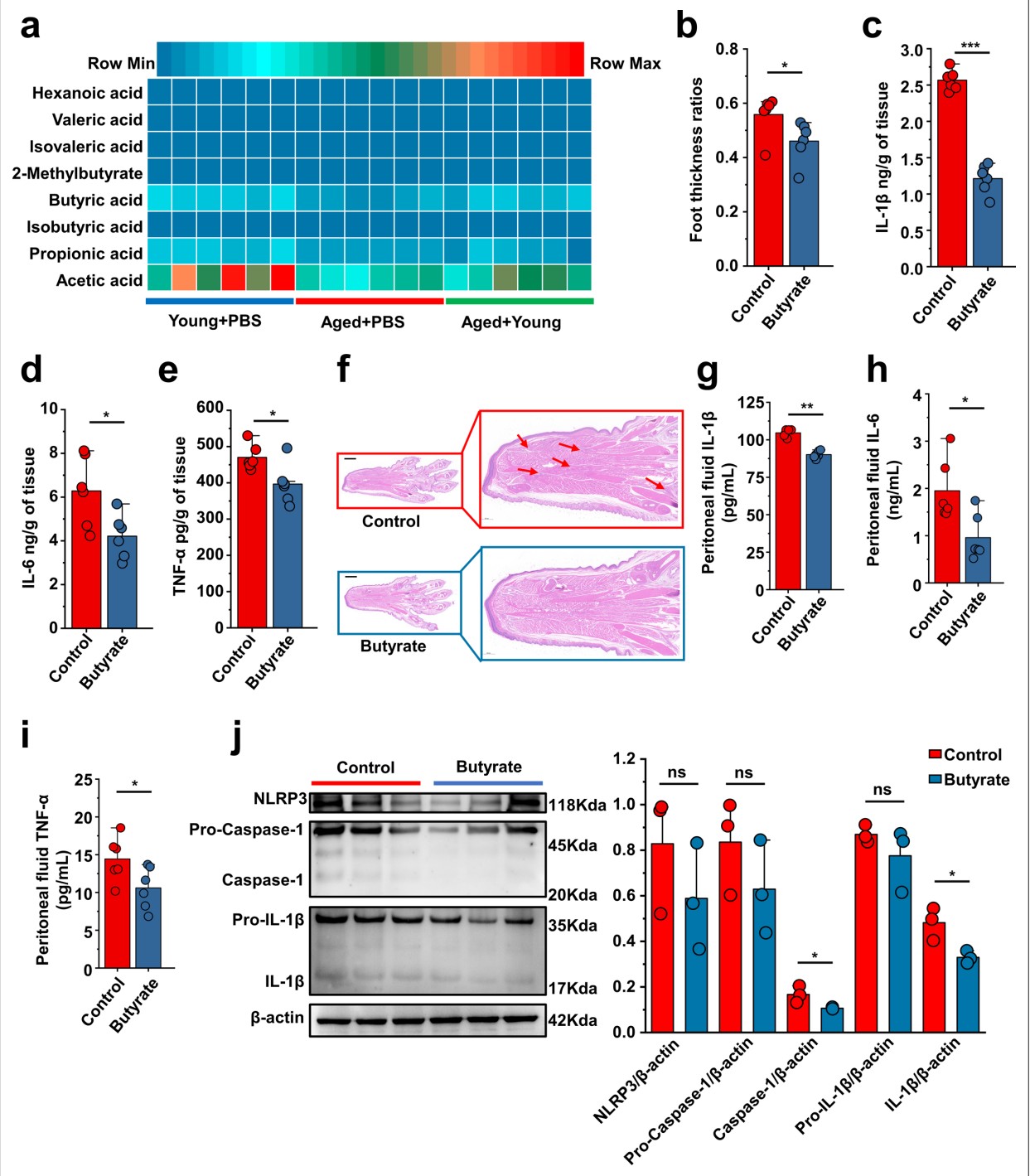

**Figure 7.** Butyrate supplementation inhibits gout and monosodium urate (MSU)-induced peritonitis. (**a**) The concentration of primary short-chain fatty acids (SCFAs) in fecal samples from both young (3 months), aged (24 months) mice, and Aged + Young (fecal microbiota transplantation [FMT] from young to aged). Graphs were generated to illustrate the changes in individual SCFAs, with a sample size of 6 in the Young + PBS, Aged + PBS, and Aged + Young group (n = 6). (**b–e**) The foot thickness ratios (**b**) were tested after MSU administration, foot tissue inflammatory parameters, including IL-1β (**c**), IL-6 (**d**), and TNF-α (**e**) concentrations, from the indicated mice are shown (n = 6). (**f**) Representative H&E-stained images of left foot tissues. Scale bars 1000 μm and ×3 magnification. (**g–i**) The peritoneal fluid concentrations of IL-1β (**g**), IL-6 (**h**), and TNF-α (**i**) inflammatory parameters were measured in the indicated mice (n = 6). (**j**) Representative western blot images of foot tissue NLRP3 pathways proteins and band density (n = 3). Values are presented as the mean ± SEM. Differences were assessed by *t*-test or one-way ANOVA and denoted as follows: *p<0.05, **p<0.01, and ***p<0.001, 'ns' indicates no significant difference between groups.

The online version of this article includes the following source data and figure supplement(s) for figure 7:

*Figure 7 continued on next page*

*Figure 7 continued*

**Source data 1.** Original Western blot image of the NLRP3 inflammasome pathway in foot tissue under butyrate.

**Source data 2.** Original Western blot image of the NLRP3 inflammasome pathway in foot tissue under butyrate.

**Figure supplement 1.** The difference analysis of the detected short-chain fatty acids and the western blot images and band density of peritoneal cells NLRP3 pathways proteins.

**Figure supplement 1—source data 1.** Original Western blot image of the NLRP3 inflammasome pathway in peritoneal cells under butyrate.

**Figure supplement 1—source data 2.** Original Western blot image of the NLRP3 inflammasome pathway in peritoneal cells under butyrate.

## Discussion

The incidence of gout and hyperuricemia continually to increases as individuals. However, research on gout among the elderly population is relatively limited. Previous studies have confirmed a close relationship between gout and the gut microbiota (*Chu et al., 2021*; *Méndez-Salazar and Martínez-Nava, 2022*), but the specific connection between the gut microbiota of older individuals and gout remains unexplored. FMT from young to old mice also reportedly alleviates age-related stroke *Lee et al., 2020* and other benefits (*Parker et al., 2022*; *Kim et al., 2022*). There are also studies indicating older patients demonstrate heightened inflammatory reactions during gout attacks (*Yang et al., 2024*). Based on this finding, we hypothesize that the gut microbiota of different age groups may exert varying effects on gout. Therefore, we conducted cross-age FMT to investigate the interaction between the gut microbiota in different age groups and gout.

This study demonstrated that FMT from young to aged mice effectively alleviated the inflammatory response caused by MSU and improved uric acid metabolism in elderly mice, reducing the symptoms of gout. In contrast, FMT from old or aged to young mice exacerbated gout, indicating the importance of the gut microbiota composition in gout development. Activation of the NLRP3 inflammasome has been implicated in the pathogenesis of gout, leading to the production of inflammatory cytokines such as IL-1β, IL-6, and TNF-α. The beneficial effects of FMT from young to aged mice could be attributed to inhibition of the NLRP3 inflammasome pathway. A 'younger' gut microbiota could suppress the activation of NLRP3, caspase-1, and IL-1β, thereby reducing the inflammatory response in gout. Furthermore, the modulation of uric acid metabolism in old or aged mice played a role in the effects of young gut microbiota transplantation. Additionally, the improvement in the intestinal tight junction integrity observed after transplantation of the gut microbiota from young to old or aged mice may contribute to enhanced elimination of uric acid. Recent studies have also shown a direct relationship between hyperuricemia and gut microbiota. Certain anaerobic microbial communities are capable of effectively degrading purines and uric acid and thereby regulate the abundance of purines in the body to maintain the body's uric acid balance (*Liu et al., 2023*; *Kasahara et al., 2023*; *Pan et al., 2020*). Our study also showed that the gut microbiota plays a role in maintaining the balance of SUA in the body. The 'aging' gut microbiota is more likely to elevate the levels of SUA in the body by suppressing the expression of uric acid excretion transporter (OAT1, OAT3). The 'younger' gut microbiota helps restrain the increase of SUA levels in old or aged mice by suppressing the expression of uric acid reabsorption transporters (URAT1) and the activity of uric acid-producing enzymes (mainly XOD) and also promotes the expression of uric acid excretion transporters (ABCG2) in aged mice. We also found that old or aged mice exhibited stronger purine metabolism (*Supplementary file 2a and b*). These results may explain the reason for the high levels of uric acid detected in elderly individuals.

Moreover, through 16S rDNA sequencing analysis, we found that the abundance of *Bifidobacterium* and *Akkermansia* was higher in young mice. Additionally, FMT from young to old or aged mice increased the abundance of the *Akkermansia* genus. There are existing reports regarding the therapeutic effects of *Bifidobacterium* (*Vieira et al., 2015*) and *Akkermansia* (*Zhang et al., 2022*) on gout and hyperuricemia. Therefore, we performed further investigation, and using microbiome-predicted functional and untargeted metabolomics data, we observed a significant enhancement in butanoate metabolism in young mice. Surprisingly, using untargeted metabolomics data, comparisons of the results after FMT from young to old or aged mice with those of old or aged mice also identified the secondary bile acid biosynthesis pathways. Studies have indicated that transplantation of the gut microbiota from healthy mice to progeroid mice can enhance the enrichment of secondary bile acid biosynthesis pathways, and the restoration of secondary bile acids may potentially contribute to extension of the healthspan and lifespan in progeroid mice (*Bárcena et al., 2019*). The current

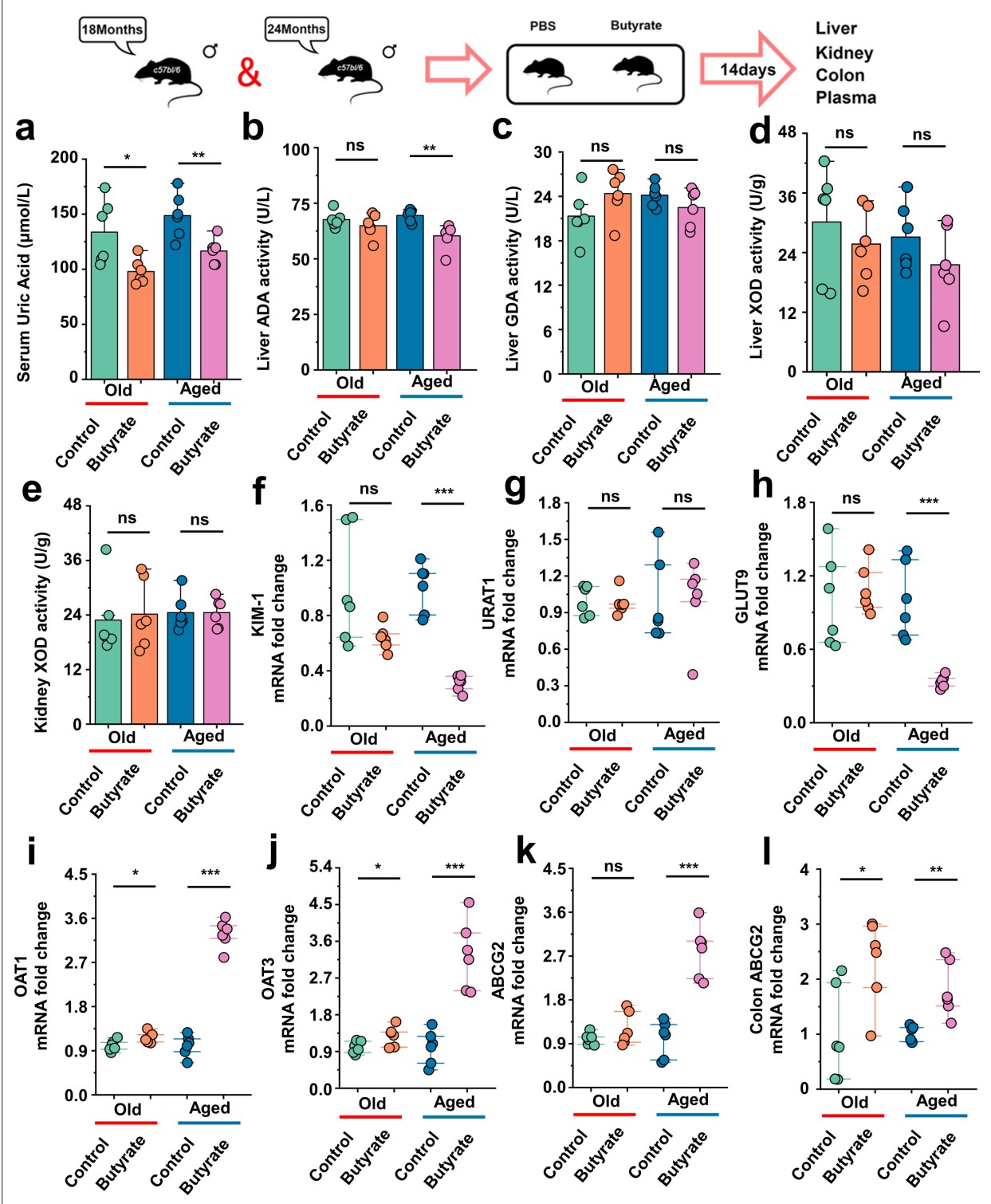

**Figure 8.** Serum uric acid-lowering effect of butyrate in old or aged mice. (**a**) The serum uric acid concentrations in Old + PBS and Old + Butyrate, Aged + PBS, and Aged + Butyrate (n = 6). (**b–d**) The activity of uric acid-producing enzymes of liver in the Old or Aged control group and the old or aged group supplemented with butyrate (n = 6), including adenosine deaminase (ADA) (**b**), guanine deaminase (GDA) (**c**), and xanthine dehydrogenase (XOD) (**d**). (**e**) The activity of XOD of kidney in the Old or Aged control group and the old or aged group supplemented with butyrate (n = 6). (**f**) Relative kidney injury molecule-1 (KIM-1) expression in the indicated groups by qPCR (n = 6). (**g, h**) Relative renal genes for uric acid reabsorption expression in the indicated groups by qPCR (n = 6), including URAT1 (**g**) and GLUT9 (**h**). (**i–k**) Relative renal genes for uric acid excretion expression in the indicated groups by qPCR (n = 6), including OAT1 (**i**), OAT3 (**j**), and ABCG2 (**k**). (**l**) Relative colonic genes for uric acid expression in the indicated groups by qPCR,

*Figure 8 continued on next page*

*Figure 8 continued*

including ABCG2 (n = 6). Values are presented as the mean ± SEM. Differences were assessed by *t*-test or one-way ANOVA and denoted as follows: *p<0.05, **p<0.01, and ***p<0.001, 'ns' indicates no significant difference between groups.

The online version of this article includes the following figure supplement(s) for figure 8:

**Figure supplement 1.** Butyrate exhibits beneficial effects on the colonic of old or aged mice.

investigation into the impact of secondary bile acids on gout and hyperuricemia is still in its incipient phase, with the intrinsic mechanisms remaining largely unelucidated, thereby rendering this domain a pivotal subject of profound scientific inquiry. Moreover, old or aged mice transplanted with gut microbiota from young mice also exhibited an increase in butanoate metabolism. Upon comparison, we discovered other bacteria genera that produce butyrate, such as *Lachnoclostridium*. Additionally, literature (*Zhao et al., 2024*; *Rios-Covian et al., 2015*) reports have indicated that *Bifidobacteria* combined with other genera can enhance the production of butyrate. Meanwhile, *Akkermansia*, particularly the species *Akkermansia muciniphila*, has been found to confer several beneficial traits, as evidenced by preclinical studies. These traits include promoting the growth of butyrate-producing bacteria through the production of acetate, which leads to a decrease in the loss of the colonic bilayer and subsequent reduction in inflammation (*Ghosh et al., 2022*). These findings led us to hypothesize that butyric acid might play a role in this phenomenon. Although *Figure 7* does show a similar trend for acetic and propionic acids as for butyric acid. However, considering the predictive data of microbial function and the non-targeted metabolomic data, there is an enhancement of Butanoate_metabolism in both young mice and elderly mice receiving young mouse intestinal microbiota transplants. Therefore, we prioritized butyrate as the subject of our study. Due to the scarcity of elderly mice, we are unable to conduct subsequent experiments with acetic and propionic acids, which is one of the limitations of this study. This work will be addressed in our follow-up research. Subsequently, we selected the Young + PBS, Aged + PBS, and Aged + Young groups for analysis of the SCFA levels and discovered that the fecal butyric acid content in young mice was higher than that in aged mice, and that FMT from young mice led to an increase in fecal butyric acid content in aged mice. Consistent with previous reports, the content of SCFAs in the feces of elderly mice was lower than that in the feces of young mice (*Yuan et al., 2024*). This result shifted our focus to butyric acid. Butyric acid, an SCFA derived from the gut microbiota, has been found to play a crucial role in host health (*Vinolo et al., 2011*). Emerging evidence indicates that among SCFAs, butyrate plays a pivotal role as a regulator in mediating metabolic control of the microbiota (*Zhang et al., 2021*). The anti-inflammatory effects of butyric acid have been extensively reported (*Li et al., 2021*; *Chen et al., 2018*), but only a few studies have investigated its impact on gout. Similarly, research on the anti-hyperuricemia effects of butyric acid is relatively scarce. We then conducted experiments using gout and peritonitis models, and found that mice supplemented with butyrate exhibited relief in symptoms of gout and inflammation. Butyrate could also reduce the SUA levels in old or aged mice by promoting the expression of uric acid excretion transporters.

The findings of this study demonstrate the positive effects of gut microbiota transplantation from young to aged mice in mitigating gout symptoms. Inhibition of the NLRP3 inflammasome pathway was found to modulate uric acid metabolism and increase butyric acid levels. The gut microbiota of young mice effectively alleviated the inflammatory response and improved uric acid metabolism in aged mice. We enhanced butanoate metabolism in the gut microbiota of young mice, while the opposite was observed in aged mice. Furthermore, we observed higher levels of butyrate in the feces of young mice. Moreover, butyrate has excellent anti-inflammatory and uric acid-lowering effects. These findings highlight the potential of targeting the gut microbiota as a therapeutic approach for the prevention and treatment of senile gout-related conditions.

## Materials and methods
### Preparation of MSU crystals

A solution consisting of 10 mM uric acid and 154 mM NaCl (both from Sigma) was adjusted to pH 7.2 and agitated for 7 days to produce MSU crystals. For sterilization, the needle-shaped crystals were

washed with ethanol, dried under sterile conditions, and heated at 180°C for 2 hr. The crystals were stored at 20 mg/mL in sterile PBS for the experimental model.

## Animals

All the mice in the different age groups were purchased from Longde Biotechnology Co., Ltd (Changchun, Jilin, China). The mice were housed in a controlled environment with a temperature of 24 ± 2°C and a 12 hr light/12 hr dark cycle in a pathogen-free facility. These mice were provided ad libitum access to food and water. All animal experiments were conducted in compliance with the regulations set forth by the Administration of Affairs Concerning Experimental Animals in China. The Institutional Animal Care and Use Committee at Jilin University approved the entire protocol, including the mouse autopsy and sample collection (20170318).

## Animal and FMT

Young (3 months), old (18 months), and aged (24 months) male mice were randomly reassigned to experimental cages 1 week prior to the experiment. The mice in the FMT groups were subjected to antibiotic interventions and depletion of the host gut microbiota by via oral gavage of a 3-day broad-spectrum antibiotic cocktail regimen (vancomycin, 100 mg/kg; metronidazole, 200 mg/kg; ampicillin 200 mg/kg, neomycin 200 mg/kg) to maximize potential engraftment of the donor microbiota. Those in the PBS groups were orally gavaged with isopycnic PBS.

Before antibiotic interventions, fecal slurries from the donor mice were prepared by pooling fecal pellet material from 12 to 15 mice per cage (chosen randomly) for each age group. At the time of pellet collection, at least three pellets from each mouse were collected into sterile centrifuge tubes, and the mice were then returned to their cage. The fecal pellets from each age group were mixed, and sterile PBS (five times the weight of the fecal pellets) was added. The homogenate was centrifuged at 500 × $g$ and 4°C for 10 min and fecal slurry was collected. After addition of 10% glycerol (v/v) and the slurry was stored at −80°C until use. After antibiotic washout, recipient mice were reassigned to different groups and received heterochronic donor microbiota twice 3 days apart by oral gavage of the fecal slurry preparation. Each mouse received 200 μL the fecal slurry preparation during each gavage.

Control groups for 3-month-old, 18-month-old, and 24-month-old receiving heterochronic transfer were gavaged with PBS only, revealed antibiotic treatment/PBS gavage only, and were administered control microbiota (obtained from a donor pool of the same age group).

Before MSU injection, fecal pellets were collected for sequencing and untargeted metabolomics analysis on day 22 and were then stored at −80°C. Blood and tissues were collected at the end of the experimental period. All FMT interventions were performed at the same time of day across all the cages to control for the circadian rhythm variabilities in feeding and the microbiota/metabolite composition.

## Experimental model details

### Gout model

To establish this model, we administered 0.4 mg of preformed MSU crystals, resuspended in 20 μL of sterile PBS via subcutaneous injection into the left hind paws of the animals. We measured the hind paw thickness using calipers (the minimum accuracy of which was 0.01 mm) before and 8 hr after MSU injection. Then, 8 hr after injection, mouse dorsal foot tissues were collected for histopathological observation and protein extraction.

### Peritonitis model

Peritonitis was induced in the mice by intraperitoneal injection of MSU crystals at a dose of 2.5 mg in PBS. Then, 6 hr later, the peritoneal cells were collected by lavage (3 mL of PBS), and the cell count was determined using a hemocytometer to ensure consistent cell numbers. The lavage fluid was used to detect cytokines, and proteins were cells extracted from the cells.

### Butyrate supplementation experiment

In this experiment, 3-month-old mice received a supplemental dose of butyrate (administered via gavage at a concentration of 200 mg/kg) for 14 days. Subsequently, these mice were subjected to

MSU stimulation. The investigation was conducted utilizing both gout and peritonitis models to assess the effects of butyrate supplementation. Simultaneously, mice aged 18 months and 24 months were given the same dosage of supplemental butyrate as the 3-month-old mice for 14 days. Subsequently, their liver, kidneys, intestines, and blood samples were collected for analysis.

## Histological analysis

The left hind paws were collected and immediately fixed with 4% paraformaldehyde for more than 48 hr. The fixed tissues were prepared after paraffin embedding. At least three slices from each tissue were prepared. The prepared sections were then dewaxed with xylene and hydrated via different gradients of alcohol. Furthermore, all hydrated sections were stained with H&E, detected under an optical microscope (Olympus, Tokyo, Japan), and analyzed with Caseviewer2.0 software.

## Western blotting

The total proteins from the peritoneal cells and foot tissue samples were extracted using a protein extraction kit (Thermo Fisher Scientific, USA). The targeted proteins were separated by 10% or 15% SDS-PAGE based on the molecular weight and then bonded to 0.22 µm PVDF membranes. After blocking with 5% skim milk for 3 hr at room temperature (RT), specific antibodies, including NLRP3, caspace-1, IL-1β, and β-actin, were used to detected target proteins at an appropriate final concentration according to the manufacturer's instructions. The PVDF membranes were then incubated with goat anti-rabbit IgG (1:20,000), washed with TBST, and determined using the ECL plus western blotting detection system (Tanon, China). The gray values of the western blotting bands were procured using Image-Pro Plus 6.0 software.

## RNA extraction and qPCR

Tissue samples were collected and total RNA was extracted using Trizol (Takara, CA, Japan) as previously described, cDNA was reverse transcribed using a TransScript Uni All-in-One (TransGen Biotech, Beijing, China), and reacted with specific primers using a FastStart Universal SYBR Green Master Mix (ROX) (Roche, Switzerland, Basel) and Bio-Rad CFX96 (Bio-Rad Laboratories, CA). The specific primers used in this study are shown in **Supplementary file 1**. The $2^{-\Delta\Delta Ct}$ method was used to calculate the relative expression levels of genes using the control group as the calibrator.

## Cytokine determination

For the detection of the IL-β, IL-6, and TNF-α concentrations in mouse foot tissue, foot tissue was harvested from different treatment groups, and 10% tissue homogenates were prepared using RIPA buffer. The prepared samples were then centrifuged at 3000 × $g$ and 4°C for 10 min, and supernatants were collected. All prepared foot tissue supernatants were determined using IL-β, IL-6, and TNF-α ELISA kits at proper dilutions according to the manufacturer's instructions (BioLegend, CA). For the detection of cytokines in serum and peritoneal fluid, samples were separated by centrifugation (500 × $g$, 4°C, 10 min) and detected using IL-β, IL-6, and TNF-α ELISA kits according to the manufacturer's instructions (BioLegend).

## Serum biochemistry analysis

The levels of SUA, serum ALT, serum AST, serum creatinine (Cr), and blood urea nitrogen (BUN) were measured using enzymatic kits from Nanjing Jiancheng Bioengineering Institute, Nanjing, China.

## Assay of uric acid synthesis enzymes

To detect the activity of ADA and GDA in the mouse liver, the liver was collected from mice in the various treatment groups and homogenized in sterile physiological saline solution (tenfold weight ratio of tissue). Following preparation, the samples were subjected to centrifugation at 3000 × $g$ and 4°C for 10 min, and the supernatants were collected. The activity of ADA and GDA in the liver supernatants was determined using ELISA kits (Feiya Biotechnology, China) according to the manufacturer's instructions with appropriate dilutions. To measure the activity of XOD in the liver and kidney of mice, tissues were collected from mice in the different treatment groups and homogenized in the buffer provided in the kit (tenfold weight ratio of tissue). The samples were then centrifuged at 8000 × $g$ and 4°C for 10 min, and the supernatants were collected. The activity of XOD in the liver and kidney

supernatants was assessed using an ELISA kit (Beijing Solarbio Science & Technology Co, Ltd, China) according to the manufacturer's instructions.

### Fecal microbiota composition analysis

For specific details, please refer to Appendix 1.

### Untargeted metabolomic analysis

For specific details, please refer to Appendix 2.

### Analysis of SCFAs in feces

For specific details, please refer to Appendix 3.

### Statistical analysis

All values were expressed as the means ± SEM. Raw data were subjected to one-way ANOVA to evaluate statistical significance between at least three groups, and pairwise comparison was conducted using Student's $t$-test. The results were considered statistically significant at $p < 0.05$.

## Acknowledgements

This work was supported by the National Key R&D Program of China (no. 2023YFD1801100), the National Natural Science Foundation of China (nos. 32202889 and 31972749), and the Natural Science Foundation of Jilin Province (no. 20220101304JC).

## Additional information

### Funding

| Funder | Grant reference number | Author |
| --- | --- | --- |
| National Natural Science Foundation of China - State Grid Corporation Joint Fund for Smart Grid | 3220889 | Naisheng Zhang |
| National Natural Science Foundation of China - State Grid Corporation Joint Fund for Smart Grid | 31972749 | Naisheng Zhang |
| Natural Science Foundation of Jilin Province | 2022010304JC | Wenlong Zhang |

The funders had no role in study design, data collection and interpretation, or the decision to submit the work for publication.

### Author contributions

Ning Song, Data curation, Software, Validation, Investigation, Visualization, Methodology, Writing – original draft; Hang Gao, Investigation, Visualization, Methodology; Jianhao Li, Validation, Investigation, Methodology; Yi Liu, Investigation; Mingze Wang, Methodology; Zhiming Ma, Resources; Naisheng Zhang, Conceptualization, Funding acquisition; Wenlong Zhang, Conceptualization, Funding acquisition, Project administration, Writing – review and editing

### Author ORCIDs

Wenlong Zhang ⓘ https://orcid.org/0000-0002-6313-2241

### Ethics

All animal experiments were conducted in compliance with the regulations set forth by the Administration of Affairs Concerning Experimental Animals in China. The Institutional Animal Care and Use

Committee at Jilin University approved the entire protocol, including the mouse autopsy and sample collection (20170318).

Reviewer #2 (Public review): https://doi.org/10.7554/eLife.98714.4.sa1
Reviewer #3 (Public review): https://doi.org/10.7554/eLife.98714.4.sa2
Author response https://doi.org/10.7554/eLife.98714.4.sa3

## Additional files

### Supplementary files
Supplementary file 1. The specific primers used in this study.

Supplementary file 2. Enhancement of butanoate metabolism contributes to the prevention of elderly gout.

MDAR checklist

### Data availability
The findings of this study have been deposited into the CNGB Sequence Archive (CNSA) of China National GenBank Database (CNGBdb, https://db.cngb.org) under the accession number CNP0004751. The original western blot images can be found on FigShare, https://doi.org/10.6084/m9.figshare.2798651.

The following datasets were generated:

| Author(s) | Year | Dataset title | Dataset URL | Database and Identifier |
| --- | --- | --- | --- | --- |
| Wenlong Z | 2024 | The original uncropped image of the immunoblot in the article | https://doi.org/10.6084/m9.figshare.27986510 | figshare, 10.6084/m9.figshare.27986510 |
| Song N | 2025 | Gut microbiota transplantation from young to aged mice mitigates senile gout | https://doi.org/10.26036/CNP0004751 | CNGB Sequence Archive, 10.26036/CNP0004751 |

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

## Appendix 1

### 16S rDNA amplicon sequencing
#### Extraction of genome DNA
The CTAB/SDS method was used to extract the total genome DNA in samples. DNA concentration and purity were monitored on 1% agarose gels. According to the concentration, DNA was diluted to 1 ng/µL with sterile water.

#### Amplicon generation
16S rRNA/18SrRNA/ITS genes in distinct regions (16S V4/16S V3/16S V3-V4/16S V4-V5, 18S V4/18S V9, ITS1/ITS2, Arc V4) were amplified with specific primer (e.g., 16S V4: 515F-806R, 18S V4: 528F-706R, 18S V9: 1380F-1510R, etc.) and barcodes. All PCR mixtures contained 15 µL of Phusion High-Fidelity PCR Master Mix (New England Biolabs), 0.2 µM of each primer and 10 ng target DNA, and cycling conditions consisted of a first denaturation step at 98°C for 1 min, followed by 30 cycles at 98°C (10 s), 50°C (30 s), and 72°C (30 s), and a final 5 min extension at 72°C.

#### PCR products quantification and qualification
Mix an equal volume of 1× loading buffer (contained SYB green) with PCR products and perform electrophoresis on 2% agarose gel for DNA detection. The PCR products were mixed in equal proportions, and then QIAGEN Gel Extraction Kit (QIAGEN, Germany) was used to purify the mixed PCR products.

#### Library preparation and sequencing
Following manufacturer's recommendations, sequencing libraries were generated with NEBNext Ultra IIDNA Library Prep Kit (Cat# E7645). The library quality was evaluated on the Qubit@ 2.0 Fluorometer (Thermo Scientific) and Agilent Bioanalyzer 2100 system. Finally, the library was sequenced on an Illumina NovaSeq platform and 250 bp paired-end reads were generated.

### Data analysis
#### Paired-end reads merged and quality control
##### 1Data split paired-end
Reads were assigned to samples based on their unique barcodes and were truncated by cutting off the barcodes and primer sequences.

##### Paired-end reads merged
Paired-end reads were merged using FLASH (version 1.2.11, http://ccb.jhu.edu/software/FLASH/)( *Magoč and Salzberg, 2011*), a very fast and accurate analysis tool designed to merge paired-end reads when at least some of the reads overlap with the reads generated from the opposite end of the same DNA fragment, and the splicing sequences were called Raw Tags.

##### Data filtration quality
Filtering on the raw tags was performed using the fastp (version 0.20.0) software to obtain high-quality Clean Tags.

##### Chimera removal
The clean tags were compared with the reference database (Silva database https://www.arb-silva.de/ for 16S/18S, Unite database https://unite.ut.ee/ for ITS) using Vsearch (version 2.15.0) to detect the chimera sequences, and then the chimera sequences were removed to obtain the Effective Tags (*Haas et al., 2011*).

### ASVs ddenoise and species annotation
#### ASVs denoise for the effective tags
Obtained previously, denoise was performed with DADA2 or deblur module in the QIIME2 software (version QIIME2-202006) to obtain initial ASVs (default: DADA2), and then ASVs with abundance less than 5 were filtered out (*Li et al., 2020*).

## Species annotation

Species annotation was performed using QIIME2 software. For 16S/18S, the annotation database is Silva database, while for ITS, it is Unite database.

## Phylogenetic relationship construction

In order to study phylogenetic relationship of each ASV and the differences of the dominant species among different samples (groups), multiple sequence alignment was performed using QIIME2 software.

## Data normalization

The absolute abundance of ASVs was normalized using a standard of sequence number corresponding to the sample with the least sequences. Subsequent analysis of alpha diversity and beta diversity was performed based on the output normalized data.

## Alpha diversity

In order to analyze the diversity, richness, and uniformity of the communities in the sample, alpha diversity was calculated from seven indices in QIIME2, including Observed_otus, Chao1, Shannon, Simpson, Dominance, Good's coverage, and Pielou_e.

Three indices were selected to identify community richness:

Observed_otus—the number of observed species (http://scikit-bio.org/docs/latest/generated/skbio.diversity.alpha.observed_otus.html)
Chao—the Chao1 estimator (http://scikit-bio.org/docs/latest/generated/skbio.diversity.alpha.chao1.html#skbio.diversity.alpha.chao)
Dominance—the Dominance index (http://scikit-bio.org/docs/latest/generated/skbio.diversity.alpha.dominance.html#skbio.diversity.alpha.dominance)

Two indices were used to identify community diversity:

Shannon—the Shannon index (http://scikit-bio.org/docs/latest/generated/skbio.diversity.alpha.shannon.html#skbio.diversity.alpha.shannon)
Simpson—the Simpson index (http://scikit-bio.org/docs/latest/generated/skbio.diversity.alpha.simpson.html#skbio.diversity.alpha.simpson)

One indice was used to calculate sequencing depth:

Coverage—the Good's coverage (http://scikit-bio.org/docs/latest/generated/skbio.diversity.alpha.goods_coverage.html#skbio.diversity.alpha.goods_coverage)

One indice was used to calculate species evenness:

Pielou_e —Pielou's evenness index (http://scikit-bio.org/docs/latest/generated/skbio.diversity.alpha.pielou_e.html#skbio.diversity.alpha.pielou_e.)

## Beta diversity

In order to evaluate the complexity of the community composition and compare the differences between samples (groups), beta diversity was calculated based on weighted and unweighted unifrac distances in QIIME2.

Cluster analysis was performed with principal component analysis (PCA), which was applied to reduce the dimension of the original variables using the ade4 package and ggplot2 package in Rsoftware (version 3.5.3).

PCoA was performed to obtain principal coordinates and visualize differences of samples in complex multi-dimensional data. A matrix of weighted or unweighted unifrac distances among samples obtained previously was transformed into a new set of orthogonal axes, where the maximum variation factor was demonstrated by the first principal coordinate, and the second maximum variation factor was demonstrated by the second principal coordinate, and so on. The three-dimensional PCoA results were displayed using QIIME2package, while the two-dimensional PCoA results were displayed using ade4 package and ggplot2package in R software (version 2.15.3).

To study the significance of the differences in community structure between groups, the adonis and anosim functions in the QIIME2 software were used to do analysis. To find out the significantly different species at each taxonomic level (phylum, class, order, family, genus, species), the R software (version 3.5.3) was used to do MetaStat and $t$-test analysis. The LEfSe software (version 1.0) was used to do LEfSe analysis (LDA score threshold: 4) to find out the biomarkers. Further, to study the functions of the communities in the samples and find out the different functions of the communities in the different groups, the PICRUSt2 software (version 2.1.2-b) was used for function annotation analysis.

## Appendix 2

### Untargeted metabolomics: Materials and methods

#### Metabolites extraction

Fecal samples (100 mg) were individually grounded with liquid nitrogen and the homogenate was resuspended with prechilled 80% methanol by well vortex. The samples were incubated on ice for 5 min and then centrifuged at 15,000 × *g*, 4°C for 20 min. Some of the supernatants were diluted to final concentration containing 53% methanol by LC-MS grade water. The samples were subsequently transferred to a fresh Eppendorf tube and then centrifuged at 15,000 × *g*, 4°C for 20 min. Finally, the supernatant was injected into the LC-MS/MS system analysis (*Want et al., 2013*).

#### UHPLC-MS/MS analysis

UHPLC-MS/MS analyses were performed using a Vanquish UHPLC system (Thermo Fisher, Germany) coupled with an Orbitrap Q Exactive TMHF-Xmass spectrometer (Thermo Fisher) at Novogene Co, Ltd (Beijing, China). Samples were injected onto a Hypesil Gold column (100 × 2.1 mm, 1.9 μm) using a 17 min linear gradient at a flow rate of 0.2 mL/min. The eluents for the positive polarity mode were eluent A (0.1% FA in water) and eluent B (methanol). The eluents for the negative polarity mode were eluent A (5 mM ammonium acetate, pH 9.0) and eluent B (methanol). The solvent gradient was set as follows: 2% B, 1.5 min; 2–85% B, 3 min; 85–100% B, 10 min; 100–2% B, 10.1 min; 2% B, 12 min. QExactive HF-X mass spectrometer was operated in positive/negative polarity mode with spray voltage of 3.5 kV, capillary temperature of 320°C, sheath gas flow rate of 35 psi, and aux gas flow rate of 10 L/min, S-lens RF level of 60, Aux gas heater temperature of 350°C.

#### Data processing and metabolite identification

The raw data files generated by UHPLC-MS/MS were processed using the Compound Discoverer 3.1 (CD3.1, Thermo Fisher) to perform peak alignment, peak picking, and quantitation for each metabolite. The main parameters were set as follows: retention time tolerance, 0.2 min; actual mass tolerance, 5 ppm; signal intensity tolerance, 30%; signal/noise ratio, 3; and minimum intensity, etc. Then, peak intensities were normalized to the total spectral intensity. The normalized data was used to predict the molecular formula based on additive ions, molecular ion peaks, and fragment ions. Then, the peaks were matched with the mzCloud (https://www.mzcloud.org/), mzVault, and MassList database to obtain the accurate qualitative and relative quantitative results. Statistical analyses were performed using the statistical software R (version R-3.4.3), Python (2.7.6 version), and CentOS (release 6.6). When data were not normally distributed, standardize according to the formula: sample raw quantitation value/(the sum of sample metabolite quantitation value/the sum of QC1 sample metabolite quantitation value) to obtain relative peak areas; and compounds whose CVs of relative peak areas in QC samples were greater than 30% were removed, and finally the metabolites' identification and relative quantification results were obtained.

#### Data analysis

These metabolites were annotated using the KEGG database (https://www.genome.jp/kegg/pathway.html), HMDB database (https://hmdb.ca/metabolites), and LIPID Maps database (http://www.lipidmaps.org/). PCA and partial least-squares discriminant analysis (PLS-DA) were performed using metaX (a flexible and comprehensive software for processing metabolomics data; *Wen et al., 2017*). We applied univariate analysis (*t*-test) to calculate the statistical significance (p-value). The metabolites with VIP > 1 and p-value<0.05 and fold change (FC) ≥2 or FC ≤ 0.5 were differential metabolites. Volcano plots were used to filter metabolites of interest which was based on $\log_2$ (foldchange) and $-\log_{10}$ (p-value) of metabolites using ggplot2 in R language.

For clustering heat maps, the data were normalized using z-scores of the intensity areas of differential metabolites and were plotted using Pheatmap package in R language. The correlation between differential metabolites was analyzed using cor () in R language (method = Pearson). Statistically significant correlation between differential metabolites was calculated using cor. mtest () in R language. p-value<0.05 was considered statistically significant, and correlation plots were plotted using corrplot package in R language. The functions of these metabolites and metabolic pathways were studied using the KEGG database. The metabolic pathways enrichment of differential metabolites was performed. When the ratio was satisfied by x/n > y/N, metabolic pathway was considered enrichment; and when the p-value of metabolic pathway <0.05, metabolic pathway was considered statistically significant enrichment.

## Appendix 3

### Reagents and instruments

#### Equipment

An ultra-high-performance liquid chromatography coupled to tandem mass spectrometry (UHPLC-MS/MS) system (Vanquish Flex UHPLC-TSQ Altis, Thermo Scientific Corp, Germany).

#### Materials and reagents

All the 11 SCFA standards were obtained from ZZ Standards Co, Ltd (Shanghai, China). Methanol (Optima LC-MS), acetonitrile (Optima LC-MS), ammonium acetate, and isopropanol (Optima LC-MS) were purchased from Thermo Fisher Scientific (FairLawn, NJ). Ultrapure water was purchased from Millipore (MA).

### Standard solution preparation

The stock solution of individual SCFA was mixed and prepared in SCFA-free matrix to obtain a series of SCFA calibrators. Certain concentrations of isotope standard were compounded and mixed as internal standard (IS). The stock solution of all of these and working solution was stored in a refrigerator at –20°C.

### Metabolites extraction

The samples (100 mg) were resuspended with liquid nitrogen and then diluted to 100 times samples. Then 100 μL was taken respectively and homogenized with 400 μL of methanol (80%) and centrifuged to remove the protein. The supernatant was added to derivatization reagent (150 μL) and derivatized at 40°C for 40 min. Then the supernatant (190 μL) was homogenized with 10 μL mixed IS solution. Finally, it was injected into the LC-MS/MS system for analysis.

### LC-MS method

A UHPLC-MS/MS system (Vanquish Flex UHPLC-TSQ Altis, Thermo Scientific Corp) was used to quantitate SCFA in Novogene Co, Ltd. Separation was performed on a Waters ACQUITY UPLC BEH C18 column (2.1 × 100 mm, 1.7 μm), which was maintained at 40°C. The mobile phase, consisting of 10 mM ammonium acetate in water (solvent A) and acetonitrile:isopropanol (1:1) (solvent B), was delivered at a flow rate of 0.30 mL/min. The solvent gradient was set as follows: initial 25% B, 2.5 min; 25–30% B, 3 min; 30–35% B, 3.5 min; 35–38% B, 4 min; 38–40% B, 4.5 min; 40–45% B, 5 min; 45–50% B, 5.5 min; 50–55% B, 6.5 min; 55–58% B, 7 min; 58–70% B, 7.5 min; 70–100% B, 7.8 min; 100–25% B, 10.1 min; and 25% B, 12 min. The mass spectrometer was operated in negative multiple reaction mode. Parameters were as follows: ion spray voltage (–4500 V), sheath gas (35 psi), ion source temperature (550°C), auxiliary gas (50 psi), and collision gas (55 psi).

