## [Editor Report · eLife Assessment]

This is an **important** study showing that age-related gut microbiota modulate uric acid metabolism through the NLRP3 inflammasome pathway and thereby regulate susceptibility to age-related gout. Several experimental approaches (mechanistic insights) and methods (data quality) remain **incomplete**. This article should be of interest to researchers working on gout and microbiota.

---

## [Referee Report · Reviewer #2 (Public review)]

Summary:

In their manuscript, the authors report that fecal transplantation from young mice into old mice alleviates susceptibility to gout. The gut microbiota in young mice is found to inhibit activation of the NLRP3 inflammasome pathway and reduce uric acid levels in the blood in the gout model.

Strengths:

The authors focused on the butanoate metabolism pathway based on the results of metabolomics analysis after fecal transplantation and identified butyrate as the key factor in mitigating gout susceptibility. In general, this is a well-performed study.

Weaknesses:

The discussion on the current results and previous studies regarding the effect of butyrate on gout symptoms is insufficient.

---

## [Referee Report · Reviewer #3 (Public review)]

The manuscript presents interesting findings on the role of gut microbiota in gout, focusing on the interplay between age-related changes, inflammation, and microbiota-derived metabolites, particularly butyrate. The study provides valuable insights into the therapeutic potential of microbiota interventions and metabolites for managing hyperuricemia and gout.

The manuscript has improved with the revisions made, particularly regarding clarifications on experimental design and the inclusion of supplementary data.

Comments on latest version:

The authors have addressed many previous concerns; however, some areas still require clarification and improvement to support more definitive conclusions.

(1) This study suggests that microbiota interventions, particularly butyrate, show promising therapeutic potential for hyperuricemia and gout. While the authors discuss the functions of certain butyrate-producing bacteria, I recommend further validating the gut microbiota-butyrate pathway by supplementing germ-free animal models with a single butyrate-producing strain, such as Clostridium butyricum. To strengthen the manuscript, I suggest the authors make further revisions to address these key issues.

(2) Additionally, I was unable to locate the full-length, uncropped Western blot images in the manuscript or supplementary materials. Could the authors please provide these?

---

## [Author Response]

The following is the authors’ response to the previous reviews.

**Reviewer #2 (Public review):**
Summary:In their manuscript the authors report that fecal transplantation from young mice into old mice alleviates susceptibility to gout. The gut microbiota in young mice is found to inhibit activation of the NLRP3 inflammasome pathway and reduce uric acid levels in the blood in the gout model.Strengths:They focused on the butanoate metabolism pathway based on the results of metabolomics analysis after fecal transplantation and identified butyrate as the key factor in mitigating gout susceptibility. In general, this is a well-performed study.Weaknesses:The discussion on the current results and previous studies regarding the effect of butyrate on gout symptoms is insufficient. The authors need to provide a more thorough discussion of other possible mechanisms and relevant literature.
**Reviewer #2 (Recommendations for the authors):**
General comments:I appreciate the authors' efforts to answer the comments raised in my previous review (as Reviewer#2). However, I still detect some issues that need to be fully addressed, with inadequate or even no answers for several comments.

Thank you for your valuable feedback. Your previous suggestions have been incredibly helpful for our paper. Although we have strived to make the article as comprehensive as possible, there may still be some areas that are not perfectly refined.

The response to comment 1: The author's statement is not very convincing. What are the trends of inflammation factors? The data in Figure 1G-H suggest that butyrate may not be the only factor to explain this phenomenon. Authors should carefully interpret the data in Figure 1G-H.

Sorry for the inadequate clarification on your question. We utilize antibiotics for treatment in order to establish the relationship between gut microbiota, age, and gout. Our research findings indicate that there is a trend for serum uric acid levels to increase with age, and we also observe that the older the age, the more pronounced the stimulation to MSU. We found that after clearing the gut microbiota and then stimulating with MSU, the trend of inflammation factors and serum uric acid level changing with age disappears. Thus, we can preliminarily draw the conclusion that the gut microbiota is closely associated with age, gout, and hyperuricemia.

The response to comment 2: I understand the importance of evaluating a range of indicators, but food thickness is the most crucial clinical marker for diagnosing goats. Please move the data from Supplemental Figure 1A to the main figure.

Thank you for your suggestions. We have included the most significant results in the main figure, and the description of “foot thickness” has already been provided descriptively in the manuscript. Additionally, considering the layout and arrangement of the images, we have placed it in the supplementary figures 1.

The response to comment 3: The immunostaining for ZO-1 and Occludin is unclear. Please provide higher magnification images to confirm the specific staining.

Thank you for your valuable feedback. We have enhanced the clarity of the images. In addition to adding immunohistochemical images in Supplementary Material 4, we have also submitted independent images.

The response to comment 4: The authors still haven't directly addressed my comment.

Please accept our sincere apologies for not providing a clearer response to your question. The indicators related to uric acid-producing enzymes and uric acid transporters have been separately analyzed according to different age groups. The specific results are detailed in section " The expression of uric acid-producing enzymes activity and uric acid transporters at the mRNA level across different age groups" of Supplementary Material 4.

No response was given for comment 5. Please address it.

In a PCoA plot, the distance between samples reflects the similarity in the structure of the microbial communities: the closer the distance, the more similar the composition of the communities; the greater the distance, the more pronounced the differences. We judge based on the relative distances of each group in the plot, observing their degree of proximity.

The response to comment 6: I understand the author's statement, and I suggest incorporating it into the discussion section of the revised manuscript.

Thank you for your suggestions. We have incorporated the relevant content into our discussion.

The response to comment 7: Again, please incorporate this statement into the discussion section of the revised manuscript.

Thank you for your suggestions. We have incorporated the relevant content into our discussion.

**Reviewer #3 (Public review):**
Summary:The revised manuscript presents interesting findings on the role of gut microbiota in gout, focusing on the interplay between age-related changes, inflammation, and microbiota-derived metabolites, particularly butyrate. The study provides valuable insights into the therapeutic potential of microbiota interventions and metabolites for managing hyperuricemia and gout. While the authors have addressed many of the previous concerns, a few areas still require clarification and improvements to strengthen the manuscript's clarity and overall impact.(1) While the authors mention that outliers in the data do not affect the conclusions, there remains a concern about the reliability of some figures (e.g., Figure 2D-G). It is recommended to provide a more detailed explanation of the statistical analysis used to handle outliers. Additionally, the clarity of the Western blot images, particularly IL-1β in Figure 3C, should be improved to ensure clear and supportive evidence for the conclusions.

Thank you for your suggestion. We respond as follows: (1) Outliers can occasionally constitute intrinsic elements of the dataset, reflecting genuine occurrences within the experimental context. The elimination of such outliers has the potential to introduce bias into the results, thereby facilitating misconceptions regarding the underlying phenomenon under investigation. In order to maintain the transparency and integrity of the dataset, we have elected to retain the outliers within our analysis. This decision is based on the recognition that these values may represent genuine experimental observations or unique conditions that are inherently meaningful to the phenomenon under investigation. By preserving these data points, we aim to provide a comprehensive and unbiased representation of the experimental results, allowing for a more nuanced interpretation of the findings. (2) Due to the scarcity of samples, we are unable to fulfill your request in the short term. Furthermore, we have noted that the band for IL-1β in Figure 3C is indeed visible and we consider it suitable for subsequent analysis.

(2) The manuscript raises a key question about why butyrate supplementation and FMT have different effects on uric acid metabolism and excretion. While the authors have addressed this by highlighting the involvement of multiple bacterial genera, it is still recommended to expand on the differences between these interventions in the discussion, providing more mechanistic insights based on available literature.

Thank you for your suggestion. We have enriched the discussion in the manuscript and included additional comparisons

(3) It is noted that IL-6 and TNF-α results in foot tissue were requested and have been added to supplementary material. However, the main text should clearly reference these additions, and the supplementary figures should be thoroughly reviewed for consistency with the main findings. The use of abbreviations (e.g., ns for no significant difference) and labeling should also be carefully checked across all figures.

Thank you for your valuable feedback. We have revised the manuscript in accordance with your suggestions.

(4) The manuscript presents butyrate as a key molecule in gout therapy, yet there are lingering concerns about its central role, especially given that other short-chain fatty acids (e.g., acetic and propionic acids) also follow similar trends. The authors should consider further acknowledging these other SCFAs and discussing their potential contribution to gout management. Additionally, the rationale for focusing primarily on butyrate in subsequent research should be made clearer.

Thank you for your input. We have incorporated additional evidence into the discussion, explaining why we ultimately chose butyrate in subsequent research.

(5) The full-length uncropped Western blot images should be provided as requested, to ensure transparency and reproducibility of the data.

Thank you for your suggestion. We have already included the relevant explanations in the manuscript.

(6) Despite the authors' revisions, several references still lack page numbers. Please ensure that all references are properly formatted, including complete page ranges.

Thank you for your suggestions; we will make more detailed revisions to the references.

The manuscript has improved with the revisions made, particularly regarding clarifications on experimental design and the inclusion of supplementary data. However, some concerns about data quality, mechanistic insights, and clarity in the figures remain. Addressing these points will enhance the overall impact of the work and its potential contribution to the understanding of the gut microbiome in gout and hyperuricemia. A final revision, with careful attention to both major and minor points, is highly recommended before resubmission.

Once again, we are grateful for your suggestions and recognition. Your input has been of immense help to our manuscript and has also provided us with a valuable learning opportunity.